# Deep Orthogonal Hypersphere Compression for Anomaly Detection

**Yunhe Zhang**[1,2]   **Yan Sun**[1,3]   **Jinyu Cai**[4]   **Jicong Fan**[1,2]*
[1]School of Data Science, The Chinese University of Hong Kong, Shenzhen, China
[2]Shenzhen Research Institute of Big Data, Shenzhen, China
[3]School of Computing, National University of Singapore, Singapore
[4]Institute of Data Science, National University of Singapore, Singapore
`zhangyhannie@gmail.com`  `yansun@comp.nus.edu.sg`
`jinyucai1995@gmail.com`  `fanjicong@cuhk.edu.cn`

## Abstract

Many well-known and effective anomaly detection methods assume that a reasonable decision boundary has a hypersphere shape, which however is difficult to obtain in practice and is not sufficiently compact, especially when the data are in high-dimensional spaces. In this paper, we first propose a novel deep anomaly detection model that improves the original hypersphere learning through an orthogonal projection layer, which ensures that the training data distribution is consistent with the hypersphere hypothesis, thereby increasing the true positive rate and decreasing the false negative rate. Moreover, we propose a bi-hypersphere compression method to obtain a hyperspherical shell that yields a more compact decision region than a hyperball, which is demonstrated theoretically and numerically. The proposed methods are not confined to common datasets such as image and tabular data, but are also extended to a more challenging but promising scenario, graph-level anomaly detection, which learns graph representation with maximum mutual information between the substructure and global structure features while exploring orthogonal single- or bi-hypersphere anomaly decision boundaries. The numerical and visualization results on benchmark datasets demonstrate the superiority of our methods in comparison to many baselines and state-of-the-art methods.

## 1 Introduction

Anomaly detection plays a crucial role in a variety of applications, including fraud detection in finance, fault detection in chemical engineering (Fan & Wang, 2014), medical diagnosis, and the identification of sudden natural disasters (Aggarwal, 2017). Significant research has been conducted on anomaly detection using both tabular and image data (Ruff et al., 2018; Fan & Chow, 2020; Goyal et al., 2020; Chen et al., 2022; Liznerski et al., 2021; Sohn et al., 2021; Liznerski et al., 2021). A common setting is to train a model solely on normal data to distinguish unusual patterns from abnormal ones, which is usually referred to as one-class classification (Schölkopf et al., 1999; Tax & Duin, 2004; Ruff et al., 2018; Pang et al., 2021; Seliya et al., 2021). For example, the support vector data description (SVDD) proposed by (Tax & Duin, 2004) obtains a spherically shaped boundary around a dataset, where data points falling outside the hypersphere will be detected as anomalous data. The deep SVDD proposed by (Ruff et al., 2018) trains a neural network to transform the input data into a space in which normal data are distributed in a hyperspherical decision region. Regarding the concern that finite training normal data generating distribution may be incomplete or draw from many sets of categories, Kirchheim et al. (2022) proposed a supervised multi-class hypersphere anomaly detection method. Han et al. (2022) provided a review and comparison of many anomaly detection methods. Compared with common anomaly detection, there is relatively little work on graph-level data, despite the fact that graph anomaly detection has application scenarios in various problems, such as identifying abnormal communities in social networks, discriminating whether human-brain networks are healthy (Lanciano et al., 2020), or detecting unusual protein structures in biological

---

*Corresponding author.

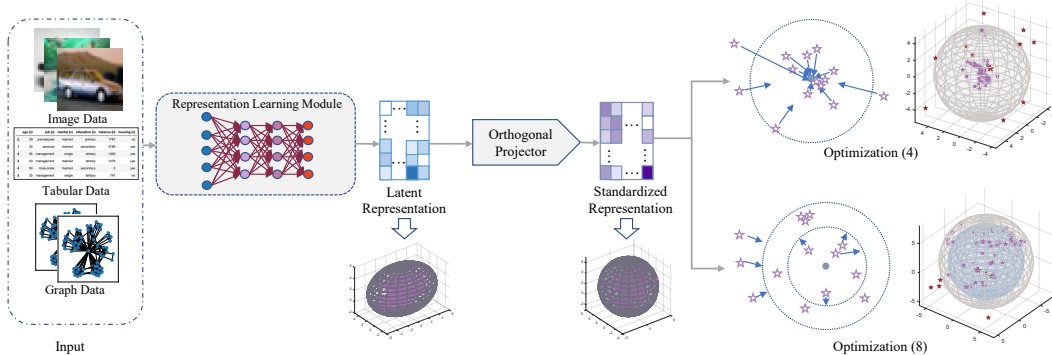

Figure 1: Architecture of the proposed models *(right top: DOHSC; right bottom: DO2HSC)*. Herein, 2-D visualizations show the trends of training data when applying two optimizations and 3-D visualizations illustrate the detection results obtained by them, respectively.

experiments. The target of graph-level anomaly detection is to explore a regular group pattern and distinguish the abnormal manifestations of the group. However, graph data are inherently complex and rich in structural and relational information. This characteristic facilitates the learning of powerful graph-level representations with discriminative patterns in many supervised tasks (e.g., graph classification) but brings many obstacles to unsupervised learning. Graph kernels (Kriege et al., 2020) are useful for both supervised and unsupervised graph learning problems. For graph-level anomaly detection, graph kernels can be combined with one-class SVM (Schölkopf et al., 1999) or SVDD (Tax & Duin, 2004). This is a two-stage approach that cannot ensure that implicit features are sufficiently expressive for learning normal data patterns. Recently, researchers proposed several end-to-end graph-level anomaly detection methods (Ma et al., 2022; Zhao & Akoglu, 2021; Qiu et al., 2022). For example, Ma et al. (2022) proposed a global and local knowledge distillation method for graph-level anomaly detection. Zhao & Akoglu (2021) combined the deep SVDD objective function and graph isomorphism network to learn a hypersphere of normal samples.

Although the hypersphere assumption is reasonable and practical, and has led to many successful algorithms (Tax & Duin, 2004; Ruff et al., 2018; 2020; Kirchheim et al., 2022; Zhao & Akoglu, 2021) for anomaly detection, it still exhibits the following three limitations:

- First, minimizing the sum of squares of the difference between each data point and the center cannot guarantee that the learned decision boundary is a standard hypersphere. Instead, one may obtain a hyperellipsoid (see Figure 2) or other shapes that are inconsistent with the assumption, which will lower the detection accuracy.

- The second is that in high-dimensional space the normal data enclosed by a hypersphere are all far away from the center (see Figure 3 and Proposition 2) with high probability. It means that there is no normal data around the center of the hypersphere; hence, the normality in the region is not supported, whereas anomalous data can still fall into the region. It's related to the *soap-bubble* phenomenon of high-dimensional statistics (Vershynin, 2018).

- Last but not least, in high-dimensional space, one hypersphere is not sufficiently compact. In other words, the distribution of normal data in the hypersphere is extremely sparse because of the high dimensionality and limited training data. A high sparsity increases the risk of detecting anomalous data as normal.

To address these issues, we propose two anomaly detection methods. The first one, **D**eep **O**rthogonal **H**yper**s**phere **C**ontraction (DOHSC), utilizes an orthogonal projection layer to render the decision region more hyperspherical and compact to reduce evaluation errors. The second one, **D**eep **O**rthogonal **Bi**-**H**yper**s**phere **C**ompression (DO2HSC), aims to solve the problem of the *soap-bubble* phenomenon and incompactness. From a 2-dimensional view, DO2HSC limits the decision area (of normal data) to an interval enclosed by two co-centered hyperspheres, and similarly learns the orthogonality-projected representation. Accordingly, a new detection metric is proposed for DO2HSC. The framework of the methods mentioned above is shown in Figure 1. In addition,

graph-level extensions of DOHSC and DO2HSC are conducted to explore a more challenging task, i.e., graph-level anomaly detection. In summary, our contributions are three-fold.

- First, we present a hypersphere contraction algorithm for anomaly detection tasks with an orthogonal projection layer to promote training data distribution close to the standard hypersphere, thus avoiding inconsistencies between assessment criteria and actual conditions.

- Second, we propose the deep orthogonal bi-hypersphere compression model to construct a decision region enclosed by two co-centered hyperspheres, which has theoretical supports and solves the problem of *soap-bubble* phenomenon and incompactness of the single-hypersphere assumption.

- Finally, we extend our methods to graph-level anomaly detection and conduct abundant experiments to show the superiority of our methods over the state-of-the-art.

## 2 DEEP ORTHOGONAL HYPERSPHERE COMPRESSION

### 2.1 VANILLA MODEL

Denote a data matrix by $\mathbf{X} \in \mathbb{R}^{n \times d}$ with $n$ instances and $d$ features, we first construct an auto-encoder and utilize the latent representation $\mathbf{Z} = f_{\mathcal{W}}^{\mathrm{enc}}(\mathbf{X})$ to initialize a decision region's center $\mathbf{c}$ according to Deep SVDD (Ruff et al., 2018), i.e, $\mathbf{c} = \frac{1}{n} \sum_{i=1}^{n} f_{\mathcal{W}}^{\mathrm{enc}}(\mathbf{x}_i)$, where $\mathbf{x}_i$ denotes the transpose of the $i$-th row of $\mathbf{X}$ and $f_{\mathcal{W}}^{\mathrm{enc}}(\cdot)$ is an $L$-layer representation learning module with parameters $\mathcal{W} = \{\mathbf{W}_l, \mathbf{b}_l\}_{l=1}^{L}$. With this center, we expect to optimize the learned representation of normal data to be distributed as close to it as possible, so that the unexpected anomalous data falling out of this hypersphere would be detected. The Hypersphere Contraction optimization problem for anomaly detection is first formulated as follows:

$$\min_{\mathcal{W}} \frac{1}{n} \sum_{i=1}^{n} \|f_{\mathcal{W}}^{\mathrm{enc}}(\mathbf{x}_i) - \mathbf{c}\|^2 + \frac{\lambda}{2} \sum_{l=1}^{L} \|\mathbf{W}_l\|_F^2, \quad (1)$$

where the regularization is to reduce over-fitting.

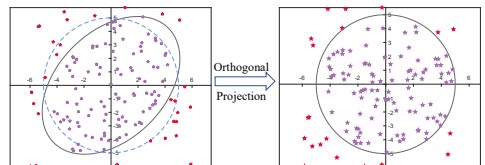

### 2.2 ORTHOGONAL PROJECTION LAYER

Although the goal of Optimization (1) is to learn a hypersphere as the decision boundary, we find that it usually yields a hyperellipsoid or even more irregular shapes (please refer to Section I in the supplementary material). This phenomenon would lead to inaccuracies in the testing stage, because the evalu-

Figure 2: Toy example of decision boundaries with and without the orthogonal projection layer. Blue circle: assumed decision boundary; black ellipse: actual decision boundary; purple points: normal data; red points: abnormal data.

ation was based on the hypersphere assumption. Figure 2 illustrates an intuitive example. In the left plot, the learned decision boundary (black ellipse) does not match the assumption (blue circle), which decreases the true-positive (TP) rate and increases the false-positive (FP) rate. Thus the detection precision, calculated as $\frac{TP\downarrow}{TP\downarrow + FP\uparrow}$, decreases compared to the right plot. The inconsistency between the assumption and the actual solution stems from the following two points: **1)** the learned features have different variances and **2)** the learned features are correlated. Clearly, these two issues cannot be avoided by solely solving Optimization (1).

To solve these issues, as shown in the right plot of Figure 2, we append an orthogonal projection layer to the feature layer, i.e., the output of $f_{\mathcal{W}}^{\mathrm{enc}}$. Note that we pursue orthogonal features of latent representation rather than computing the projection onto the column or row space of $\mathbf{Z} \in \mathbb{R}^{n \times k}$, which is equivalent to performing Principal Component Analysis (PCA) (Wold et al., 1987) and using standardized principal components. Our experiments also justify the necessity of this projection step and the standardization process, which will be discussed further in Appendix K. Specifically, the projection layer is formulated as

$$\tilde{\mathbf{Z}} = \mathrm{Proj}_{\Theta}(\mathbf{Z}) = \mathbf{Z}\mathbf{W}^*, \quad \text{subject to } \tilde{\mathbf{Z}}^{\top}\tilde{\mathbf{Z}} = \mathbf{I}_{k'} \quad (2)$$

where $\Theta := \{\mathbf{W}^* \in \mathbb{R}^{k \times k'}\}$ is the set of projection parameters, $\mathbf{I}_{k'}$ denotes an identity matrix, and $k'$ is the projected dimension. To achieve (2), one may consider adding a regularization term $\frac{\alpha}{2}\|\tilde{\mathbf{Z}}^\top \tilde{\mathbf{Z}} - \mathbf{I}_{k'}\|_F^2$ with large enough $\alpha$ to the objective, which is not very effective and will lead to one more tuning hyperparameter. Instead, we propose to achieve (2) via singular value decomposition:

$$\mathbf{U}\mathbf{\Lambda}\mathbf{V}^\top = \mathbf{Z}, \quad \mathbf{W} := \mathbf{V}_{k'}\mathbf{\Lambda}_{k'}^{-1}. \tag{3}$$

Assume that there are $b$ samples in one batch, $\mathbf{\Lambda} = \mathrm{diag}(\rho_1, \rho_2, ..., \rho_b)$ and $\mathbf{V}$ are the diagonal matrix with singular values and right-singular matrix of $\mathbf{Z}$, respectively. It is noteworthy that $\mathbf{V}_{k'} := [\mathbf{v}_1, ..., \mathbf{v}_{k'}]$ denotes the first $k'$ right singular vectors, and $\mathbf{\Lambda}_{k'} := \mathrm{diag}(\rho_1, ..., \rho_{k'})$. In each forward propagation epoch, the original weight parameter is substituted into a new matrix $\mathbf{W}^*$ in the subsequent loss computations.

## 2.3 ANOMALY DETECTION

Attaching with an orthogonal projection layer, the improved initialization of the center is rewritten in the following form $\tilde{\mathbf{c}} = \frac{1}{n}\sum_{i=1}^n \tilde{\mathbf{z}}_i$, which will be **fixed** until optimization is completed. The final objective function for anomaly detection tasks in a mini-batch would become

$$\min_{\Theta, \mathcal{W}} \frac{1}{b}\sum_{i=1}^b \|\tilde{\mathbf{z}}_i - \tilde{\mathbf{c}}\|^2 + \frac{\lambda}{2}\sum_{\mathbf{W} \in \mathcal{W}} \|\mathbf{W}\|_F^2. \tag{4}$$

After the training stage, the decision boundary $\hat{r}$ will be **fixed**, which is calculated based on the $1-\nu$ percentile of the training data distance distribution:

$$\hat{r} = \arg\min_r \mathcal{P}(\mathbf{D} \leq r) \geq \nu \tag{5}$$

where $\mathbf{D} := \{d_i\}_{i=1}^N$ follows a sampled distribution $\mathcal{P}$, and $d_i = \|\tilde{\mathbf{z}}_i - \tilde{\mathbf{c}}\|$. Accordingly, the anomalous score of $i$-th instance is defined as follows:

$$s_i = d_i^2 - \hat{r}^2 \tag{6}$$

where $\mathbf{s} = (s_1, s_2, \ldots, s_n)$. It is evident that when the score is positive, the instance is identified as abnormal, and the opposite is considered normal.

The detailed procedures are summarized in Algorithm 1 (see Appendix A), which is termed as DOHSC. DOHSC is easy to implement and can ensure that the actual decision boundary is close to a hypersphere. Our numerical results in Section 4 will show the effectiveness.

# 3 DEEP ORTHOGONAL BI-HYPERSPHERE COMPRESSION

## 3.1 MOTIVATION AND THEORETICAL ANALYSIS

As mentioned in the third paragraph of Section 1, the hypersphere assumption may encounter the soap-bubble phenomenon and incompactness. They can be succinctly summarized as

- High-dimensional data enclosed by a hypersphere are far from the center naturally, which means the normality within a wide range of distance is not supported.

- In high-dimensional space, the data distribution within a hypersphere is highly sparse, which leads to an incompact decision region and, hence, a heightened risk of detecting abnormal data as normal.

In this section, we present a detailed analysis. Let the anomaly score be determined using $\|\mathbf{z} - \mathbf{c}\|$ where $\mathbf{c}$ denotes the centroid. The original evaluation of anomaly detection compares the score with a threshold $\hat{r}$ determined by a certain quantile (e.g., 0.95). Specifically, if $\|\mathbf{z} - \mathbf{c}\| \geq \hat{r}$, $\mathbf{z}$ is abnormal. This target promoted the location of most samples near the origin. However, empirical exploration has found that most samples are far away from their origin in a high-dimensional space. Taking Gaussian distributions as an example, the distributions would look like a *soap-bubble*[1], which

---

[1]https://www.inference.vc/high-dimensional-gaussian-distributions-are-soap-bubble/

means the high-dimensional normal data may be more likely to locate in the interval region of bi-hypersphere instead of a simple hypersphere. Vershynin (2018) stated that the typical set, where data has information closest to the expected entropy of the population, of a Gaussian is the thin shell within a distance from the origin, just like the circumstances shown in Figure 3. The higher the dimensionality of the data, the more sampled instances are from the center. We also supplement the anomaly detection simulation of high-dimensional Gaussian data in Appendix C to show the significant meaning of bi-hypersphere learning. This is formally proven by the following proposition (derived from Lemma 1 of (Laurent & Massart, 2000)):

**Proposition 1.** *Suppose* $\mathbf{z}_1, \mathbf{z}_2, \cdots, \mathbf{z}_n$ *are sampled from* $\mathcal{N}(\mathbf{0}, \mathbf{I}_d)$ *independently. Then, for any* $\mathbf{z}_i$ *and all* $t \geq 0$*, the following inequality holds.*

$$\mathbb{P}\left[\|\mathbf{z}_i\| \geq \sqrt{d - 2\sqrt{dt}}\right] \geq 1 - e^{-t}.$$

The proposition shows that when the dimension is high, each $\mathbf{z}_i$ is outside the hypersphere of radius $r' := \sqrt{d - 2\sqrt{dt}}$ with a probability of at least $1 - e^{-t}$. When $r'$ is closer to $\hat{r}$ (refer to equation 5), normal data are more likely to be away from the center (see Figure 3).

Note that, in anomaly detection, $\tilde{\mathbf{z}}_i$ (e.g., the learned latent representation) is not necessarily an isotropic Gaussian. However, we obtain the following result.

**Proposition 2.** *Let* $\mathbf{z}_i = \tilde{\mathbf{z}}_i$, $i = 1, \ldots, N$ *and let* $f : \mathbb{R}^k \to \mathbb{R}^k$ *be an* $\eta$*-Lipschitz function such that* $\mathbf{s} = f(\mathbf{z})$ *are isotropic Gaussian* $\mathcal{N}(\bar{\mathbf{c}}, \mathbf{I}_k)$*. Let* $\mathbf{c}$ *be a predefined center of* $\{\mathbf{z}_i\}_{i=1}^N$ *and suppose* $\|\bar{\mathbf{c}} - f(\mathbf{c})\| \leq \epsilon$*. Then for any* $\mathbf{z}_i$ *and all* $t \geq 0$*, the following inequality holds:*

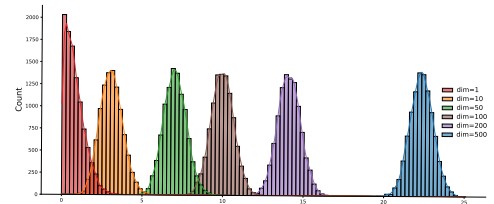

Figure 3: *Soap-bubble* phenomenon showed by the histogram of distances from the center of $10^4$ samples drawn from $\mathcal{N}(\mathbf{0}, \mathbf{I}_d)$. In high-dimensional space, almost all data are far from the center.

$$\mathbb{P}\left[\|\mathbf{z}_i - \mathbf{c}\| \geq \eta^{-1}\left(\sqrt{k - 2\sqrt{kt}} - \epsilon\right)\right] \geq 1 - e^{-t}.$$

The proposition (proved in Appendix D) indicates that most data $(N')$ satisfy $\|\mathbf{z} - \mathbf{c}\| \geq r' := \eta^{-1}\left(\sqrt{k - 2\sqrt{kt}} - \epsilon\right)$ with a probability of approximately $\binom{N'}{N}(1 - e^{-t})^{N'} e^{-t(N-N')}$, where $r'$ is close to $\hat{r}$. This means there is almost no normal training data within the range $[0, r']$, i.e. normality within the range is not supported by the normal training data. An intuitive example is:

**Example 1.** *Assume the hypersphere is centered at the origin. Consider a data point with all features very close or even equal to zero. This point is very different from the normal training data and should be abnormal data. However, according to the metric* $\|\tilde{\mathbf{z}} - \tilde{\mathbf{c}}\|$*, this point is still in the hypersphere and is finally detected as normal data.*

Given the implications of Proposition 2, we recognize that in high-dimensional spaces, traditional distance-to-center based anomaly scores (equation 6) may lose their reliability due to the concentration of measure phenomenon. Figure 4 shows a real example of abnormal data falling into a region close to the center of the hypersphere. In addition to the *soap-bubble* phenomenon, we claim that the data distribution in a high-dimensional sphere is very sparse when the number $n$ of the training data is limited. This means that when $n$ is not sufficiently large, there could be large empty holes or regions in which normality is not supported because of the randomness. It is not sufficient to treat data that fall into holes or regions as normal data. Intuitively, for example, the distribution of $n$ random points in a 3-D sphere of radius $r$ is much sparser than that in a 2-D circle of radius $r$. More formally, in a hypersphere of radius $r$ in $k$-dimensional space, the expected number of data points per unit volume is $\varrho_k = \frac{n\Gamma(k/2+1)}{\pi^{k/2}r^k}$, where $\Gamma$ is Euler's gamma function. When $r$ is not too small, $\varrho_k$ increases rapidly as $k$ decreases. See below.

**Example 2.** *Suppose* $n = 1000$*,* $r = 5$*. Then* $\varrho_2 \approx 12.7$*,* $\varrho_5 \approx 0.06$*, and* $\varrho_{10} < 0.0001$*.*

We hope to construct a more compact decision region, one with a much larger $\varrho_k$, without changing the feature dimensions.

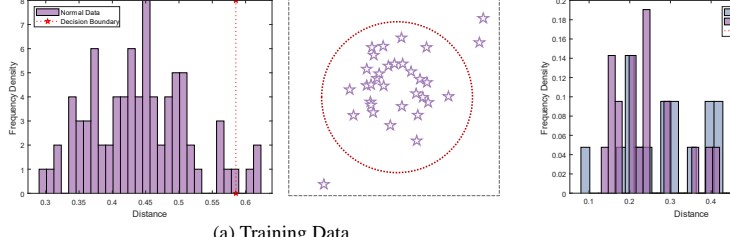

(a) Training Data          (b) Testing Data

Figure 4: Illustration of inevitable flaws in DOHSC on both the training and testing data of COX2. Left: the $\ell_2$-norm distribution of 4-dimensional distances learned from the real dataset; Right: the pseudo-layout in two-dimensional space sketched by reference to the empirical distribution.

## 3.2 ARCHITECTURE OF DO2HSC

To solve the issues discussed in the previous section, we propose an improved approach, DO2HSC, which sets the decision boundary as an interval region between two co-centered hyperspheres. This can narrow the scope of the decision area to induce normal data to fill as much of the entire interval area as possible.

After the same representation learning stage, we first utilize the DOHSC model for a few epochs to initialize the large radius $r_{\max}$ and small radius $r_{\min}$ of the interval area according to the $1 - \nu$ percentile and $\nu$ of the sample distance distribution, respectively. The aforementioned descriptions can be mathematically denoted as follows:

$$r_{\max} = \arg\min_{r} \mathcal{P}(\mathbf{D} \leq r) \geq \nu, \quad r_{\min} = \arg\min_{r} \mathcal{P}(\mathbf{D} \leq r) \geq 1 - \nu. \tag{7}$$

After fixing $r_{\max}$ and $r_{\min}$, the objective function of DO2HSC is formulated as follows:

$$\min_{\Theta, \mathcal{W}} \frac{1}{b} \sum_{i=1}^{b} \left( \max\{d_i, r_{\max}\} - \min\{d_i, r_{\min}\} \right) + \frac{\lambda}{2} \sum_{\mathbf{W} \in \mathcal{W}} \|\mathbf{W}\|_F^2. \tag{8}$$

This decision loss has the lowest bound $r_{\max} - r_{\min}$. In addition, the evaluation standard of the test data must also be changed based on this interval structure. Specifically, all instances located in the inner hypersphere and outside the outer hypersphere should be identified as anomalous individuals; only those located in the interval area should be regarded as normal data. We reset a new score function to award the positive samples beyond $[r_{\min}, r_{\max}]$ while punishing the negative samples within this range. Accordingly, the distinctive scores are calculated by

$$s_i = (d_i - r_{\max}) \cdot (d_i - r_{\min}), \tag{9}$$

where $i \in \{1, ..., n\}$. In this manner, we can also effectively identify a sample's abnormality based on its score. In general, an improved deep anomaly detection algorithm changes the decision boundary and makes the normal area more compact. Furthermore, a new practical evaluation was proposed to adapt to the improved detection method. Finally, we summarize the detailed optimization procedures in Algorithm 2 (see Appendix A).

The following proposition justifies the superiority of bi-hypersphere compression over single-hypersphere contraction from another perspective:

**Proposition 3.** *Suppose the number of normal training data is $n$, the radius of the hypersphere given by DOHSC is $r_{max}$, and the radii of the hyperspheres given by DO2HSC are $r_{max}$ and $r_{min}$ respectively. Without loss of generality, assume that all the training data are included in the learned decision regions. The ratio between the support densities of the decision regions given by DO2HSC and DOHSC is $\kappa = \frac{1}{1 - (r_{min}/r_{max})^k}$.*

In the proposition (proved in Appendix E), density is defined as the number of normal data in unit volume. A higher density indicates a higher confidence in treating a data point falling into the decision region as normal data, or treating a data point falling outside the decision region as anomalous data. Because $\kappa > 1$, the DO2HSC provides a more reliable decision region than the DOHSC. The advantage of the DO2HSC over the DOHSC is more significant when $k$ is smaller or $r_{\min}$ is closer to $r_{\max}$. Here are some examples.

**Example 3.** *Suppose $k = 50$. When $r_{min}/r_{max} = 0.9$, $\kappa \approx 1.01$. When $r_{min}/r_{max} = 0.99$, $\kappa \approx 2.5$. Suppose $k = 10$. When $r_{min}/r_{max} = 0.9$, $\kappa \approx 1.5$. When $r_{min}/r_{max} = 0.99$, $\kappa \approx 10.5$.*

### 3.3 GENERALIZATION TO GRAPH-LEVEL ANOMALY DETECTION

Given a set of graphs $\mathbb{G} = \{G_1, ..., G_N\}$ with $N$ samples, the proposed model aims to learn a $k$-dimensional representation and then set a soft boundary accordingly. In this paper, the Graph Isomorphism Network (GIN) (Xu et al., 2019) is employed to obtain the graph representation in three stages: first, input the graph data and integrate neighbors of the current node (AGGREGATE); second, combine neighbor and current node features (CONCAT); and finally, all node information (READOUT) is integrated into one global representation. Mathematically, the $i$-th node features of $l$-th layer and the global features of its affiliated $j$-th graph are denoted by

$$\mathbf{z}_\Phi^i = \text{CONCAT}(\{\mathbf{z}_i^{(l)}\}_{l=1}^L), \quad \mathbf{Z}_\Phi(G_j) = \text{READOUT}(\{\mathbf{z}_\Phi^i\}_{i=1}^{|G_j|}), \tag{10}$$

where $\mathbf{z}_\Phi^i \in \mathbb{R}^{1 \times k}$ and $\mathbf{Z}_\Phi(G_j) \in \mathbb{R}^{1 \times k}$. To integrate the contained information and enhance the differentiation between node- and global-level representations, we append additional fully connected layers denoted by the forms $M_\Upsilon(\cdot)$ and $T_\Psi(\cdot)$, respectively, where $\Upsilon$ and $\Psi$ are the parameters of the added layers. So the integrated node-level and graph-level representations are

$$\mathbf{h}_{\Phi,\Upsilon}^i := M_\Upsilon(\mathbf{z}_\Phi^i); \quad \mathbf{H}_{\Phi,\Psi}(G_j) := T_\Psi(\mathbf{Z}_\Phi(G_j)). \tag{11}$$

To better capture the local information, we utilize the batch optimization property of neural networks to maximize the mutual information (MI) between local and global representations in each batch $\mathbf{G} \subseteq \mathbb{G}$, which is defined by (Sun et al., 2020) as follows:

$$\hat{\Phi}, \hat{\Psi}, \hat{\Upsilon} = \arg\max_{\Phi,\Psi,\Upsilon} I_{\Phi,\Psi,\Upsilon}\left(\mathbf{h}_{\Phi,\Upsilon}, \mathbf{H}_{\Phi,\Psi}(\mathbf{G})\right). \tag{12}$$

Specifically, the mutual information estimator $I_{\Phi,\Psi,\Upsilon}$ follows the Jensen-Shannon MI estimator (Nowozin et al., 2016) with a positive-negative sampling method, as follows:

$$I_{\Phi,\Psi,\Upsilon}\left(\mathbf{h}_{\Phi,\Upsilon}, \mathbf{H}_{\Phi,\Psi}(\mathbf{G})\right) := \sum_{G_j \in \mathbf{G}} \frac{1}{|G_j|} \sum_{u \in G_j} I_{\Phi,\Psi,\Upsilon}\left(\mathbf{h}_{\Phi,\Upsilon}^u(G_j), \mathbf{H}_{\Phi,\Psi}(\mathbf{G})\right)$$
$$= \sum_{G_j \in \mathbf{G}} \frac{1}{|G_j|} \sum_{u \in G_j} \left[\mathbb{E}\left(-\sigma\left(-\mathbf{h}_{\Phi,\Upsilon}^u(\mathbf{x}^+) \times \mathbf{H}_{\Phi,\Psi}(\mathbf{x})\right)\right) - \mathbb{E}\left(\sigma\left(\mathbf{h}_{\Phi,\Upsilon}^u(\mathbf{x}^-) \times \mathbf{H}_{\Phi,\Psi}(\mathbf{x})\right)\right)\right], \tag{13}$$

where $\sigma(z) = \log(1 + e^z)$. For $\mathbf{x}$ as an input sample graph, we calculate the expected mutual information using its positive samples $\mathbf{x}^+$ and negative samples $\mathbf{x}^-$, which are generated from the distribution across all graphs in a subset. Given that $G = (\mathcal{V}_G, \mathcal{E}_G)$ and the node set $\mathcal{V}_G = \{v_i\}_{i=1}^{|G|}$, the positive and negative samples are divided in this manner: $\mathbf{x}^+ = \mathbf{x}_{ij}$ if $v_i \in G_j$ otherwise, $\mathbf{x}^+ = 0$. Additionally, $\mathbf{x}^-$ produces the opposite result for each of the above conditions. Thus, a data-enclosing decision boundary is required for our anomaly detection task. Let $\tilde{\mathbf{H}}_{\Phi,\Psi,\Theta}(G) = \text{Proj}_\Theta(\mathbf{H}_{\Phi,\Psi}(G))$, the center of this decision boundary should be initialized through

$$\tilde{\mathbf{c}} = \frac{1}{N} \sum_{i=1}^N \tilde{\mathbf{H}}_{\Phi,\Psi,\Theta}(G_i). \tag{14}$$

Collectively, the weight parameters of $\Phi$, $\Psi$ and $\Upsilon$ are $\mathcal{Q} := \Phi \cup \Psi \cup \Upsilon$, and let $\mathcal{R}(Q) = \sum_{\mathbf{Q} \in \mathcal{Q}} \|\mathbf{Q}\|_F^2$, we formulate the objective function of the graph-level DOHSC as

$$\min_{\Theta,\Phi,\Psi,\Upsilon} \frac{1}{|\mathbf{G}|} \sum_{i=1}^{|\mathbf{G}|} \|\tilde{\mathbf{H}}_{\Phi,\Psi,\Theta}(G_i) - \tilde{\mathbf{c}}\|^2 - \lambda \sum_{\mathbf{G} \in \mathbb{G}} I_{\Phi,\Psi,\Upsilon}\left(\mathbf{h}_{\Phi,\Upsilon}, \tilde{\mathbf{H}}_{\Phi,\Psi,\Theta}(\mathbf{G})\right) + \frac{\mu}{2}\mathcal{R}(Q), \tag{15}$$

where $|\mathbf{G}|$ denotes the number of graphs in batch $\mathbf{G}$ and $\lambda$ is a trade-off factor, the third term is a network weight decay regularizer with the hyperparameter $\mu$. Correspondingly, the objective function of graph-level DO2HSC is

$$\min_{\Theta,\Phi,\Psi,\Upsilon} \frac{1}{|\mathbf{G}|} \sum_{i=1}^{|\mathbf{G}|} (\max\{d_i, r_{\max}\} - \min\{d_i, r_{\min}\}) - \lambda \sum_{\mathbf{G} \in \mathbb{G}} I_{\Phi,\Psi,\Upsilon}\left(\mathbf{h}_{\Phi,\Upsilon}, \tilde{\mathbf{H}}_{\Phi,\Psi,\Theta}(\mathbf{G})\right) + \frac{\mu}{2}\mathcal{R}(Q). \tag{16}$$

## 4 NUMERICAL RESULTS

### 4.1 EXPERIMENTS ON IMAGE DATA

**Datasets:** Two image datasets (Fashion-MNIST, CIFAR-10) are chosen to conduct this experiment. Please refer to the detailed statistic descriptions in Appendix F.

**Baselines:** We followed the settings in (Ruff et al., 2018) and utilized the Area Under Operating Characteristic Curve (AUC) of several state-of-the-art anomaly detection algorithms, including Deep SVDD (Ruff et al., 2018), OCGAN (Perera et al., 2019), HRN-L2 and HRN (Hu et al., 2020), PLAD (Cai & Fan, 2022), and DROCC (Goyal et al., 2020). All SOTAs' results are given according to their officially reported results or are reproduced by official codes.

Table 1: Average AUCs (%) in one-class anomaly detection on CIFAR-10. * denotes we run the official released code to obtain the results, and the top two results are marked in **bold**.

| Normal Class | Airplane | Auto Mobile | Bird | Cat | Deer | Dog | Frog | Horse | Ship | Truck |
|---|---|---|---|---|---|---|---|---|---|---|
| Deep SVDD | 61.7 | 65.9 | 50.8 | 59.1 | 60.9 | 65.7 | 67.7 | 67.3 | 75.9 | 73.1 |
| OCGAN | 75.7 | 53.1 | 64.0 | 62.0 | 72.3 | 62.0 | 72.3 | 57.5 | 82.0 | 55.4 |
| DROCC* | **82.1** | 64.8 | 69.2 | 64.4 | 72.8 | 66.5 | 68.6 | 67.5 | 79.3 | 60.6 |
| HRN-L2 | 80.6 | 48.2 | 64.9 | 57.4 | **73.3** | 61.0 | 74.1 | 55.5 | 79.9 | 71.6 |
| HRN | 77.3 | 69.9 | 60.6 | 64.4 | 71.5 | 67.4 | 77.4 | 64.9 | 82.5 | 77.3 |
| PLAD | **82.5** | 80.8 | 68.8 | 65.2 | 71.6 | 71.2 | 76.4 | 73.5 | 80.6 | 80.5 |
| DOHSC | 80.3 (0.0) | **81.0** (0.0) | 70.4 (1.9) | 68.0 (1.8) | 72.1 (0.0) | 72.4 (2.1) | 83.1 (0.0) | 74.1 (0.4) | 83.3 (0.7) | 81.1 (0.7) |
| DO2HSC | 81.3 (0.2) | 82.7 (0.3) | 71.3 (0.4) | 71.2 (1.3) | 72.9 (2.1) | 72.8 (0.2) | 83.0 (0.6) | 75.5 (0.4) | 84.4 (0.5) | 82.0 (0.9) |

Considering that there is not much room for performance improvement on Fashion-MNIST, we only reproduced the results of recent or most relative algorithms, which contains Deep SVDD (Ruff et al., 2018), and DROCC (Goyal et al., 2020). The network architecture of Deep SVDD is set the same as ours for fairness.

**Results:** The experimental results are listed in Table 1. On CIFAR-10, both DOHSC and DO2HSC surpassed SOTAs, especially for Dog and Frog. Second, DO2HSC obtained better results compared with DOHSC, which further verifies the effectiveness of bi-hypersphere anomaly detection and fully demonstrates its applicability to image data. It is also worth mentioning that Deep SVDD plays an important baseline role relative to DOHSC, and DOHSC outperforms it by a large margin in all classes. This illustrates the significant meaning of the proposed orthogonal projection method is constructive. The result of Fashion-MNIST is in Appendix G.

### 4.2 EXPERIMENTS ON TABULAR DATA

**Datasets:** Here, we use two tabular datasets (Thyroid, Arrhythmia), and we followed the data split settings in Zong et al. (2018).

**Results:** The F1-scores of our methods and six baselines are reported in Table 2. A significant margin was observed between the baselines and ours, especially the results of Thyroid. Despite the challenge posed by the small sample size of the Arrhythmia data, DO2HSC still outperforms PLAD by a margin of 3%. Similarly, the orthogonal projection of DOHSC successfully standardized the results of Deep SVDD.

Table 2: Average F1-scores with the standard deviation in one-class anomaly detection on two tabular datasets. The best two results are marked in **bold**.

| | Thyroid | Arrhythmia |
|---|---|---|
| OCSVM (Schölkopf et al., 1999) | 0.56 ± 0.01 | 0.64 ± 0.01 |
| Deep SVDD (Ruff et al., 2018) | 0.73 ± 0.00 | 0.54 ± 0.01 |
| LOF (Breunig et al., 2000) | 0.54 ± 0.01 | 0.51 ± 0.01 |
| GOAD (Bergman & Hoshen, 2020) | 0.75 ± 0.01 | 0.52 ± 0.02 |
| DROCC (Goyal et al., 2020) | 0.78 ± 0.03 | 0.69 ± 0.02 |
| PLAD (Cai & Fan, 2022) | 0.77 ± 0.01 | **0.71 ± 0.02** |
| DOHSC | **0.92 ± 0.01** | 0.70 ± 0.03 |
| DO2HSC | **0.98 ± 0.59** | **0.74 ± 0.02** |

### 4.3 EXPERIMENTS ON GRAPH DATA

**Datasets:** We further evaluate our models on six real-world graph datasets[2] (COLLAB, COX2, ER_MD, MUTAG, DD and IMDB-Binary). Our experiments followed the standard one-class settings and data-split method in a previous work (Zhao & Akoglu, 2021; Qiu et al., 2022).

**Baselines:** We compare our methods with the following methods, including four graph kernels combined with OCSVM and four state-of-the-art baselines: RW (Gärtner et al., 2003; Kashima et al., 2003), SP (Borgwardt & Kriegel, 2005), WL (Shervashidze et al., 2011) and NH (Hido &

---

[2]https://ls11-www.cs.tu-dortmund.de/staff/morris/graphkerneldatasets

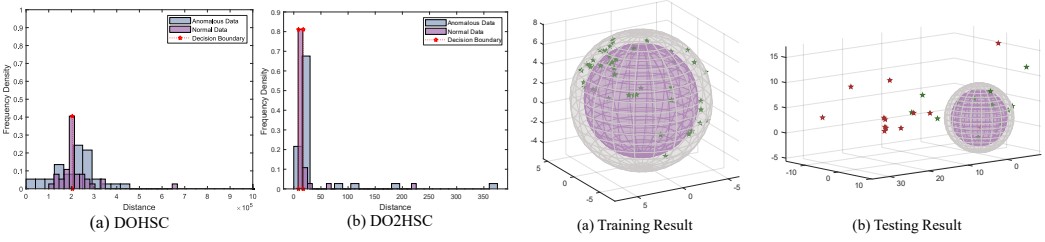

Figure 5: Distance Histograms on ER_MD.  Figure 6: 3-D plots of DO2HSC on MUTAG.

Table 3: Average AUCs with standard deviation (10 trials) of different graph-level anomaly detection algorithms. 'DSVDD' stands for 'Deep SVDD'. We assess models by regarding every data class as normal data, respectively. The best two results are highlighted in **bold** and '–' means out of memory.

| | COLLAB | | | MUTAG | | ER_MD | |
|---|---|---|---|---|---|---|---|
| | 0 | 1 | 2 | 0 | 1 | 0 | 1 |
| SP+OCSVM | $0.5910 \pm 0.0000$ | $0.8397 \pm 0.0000$ | $0.7902 \pm 0.0000$ | $0.5917 \pm 0.0000$ | $0.2608 \pm 0.0000$ | $0.4092 \pm 0.0000$ | $0.3824 \pm 0.0000$ |
| WL+OCSVM | $0.5122 \pm 0.0000$ | $0.8054 \pm 0.0000$ | $0.7996 \pm 0.0000$ | $0.6509 \pm 0.0000$ | $0.2960 \pm 0.0000$ | $0.4571 \pm 0.0000$ | $0.3262 \pm 0.0000$ |
| NH+OCSVM | $0.5976 \pm 0.0000$ | $0.8054 \pm 0.0000$ | $0.6414 \pm 0.0000$ | $0.7959 \pm 0.0274$ | $0.1679 \pm 0.0062$ | $0.5155 \pm 0.0200$ | $0.3648 \pm 0.0000$ |
| RW+OCSVM | – | – | – | $0.8698 \pm 0.0000$ | $0.1504 \pm 0.0000$ | $0.4820 \pm 0.0000$ | $0.3484 \pm 0.0000$ |
| OCGIN | $0.4217 \pm 0.0606$ | $0.7565 \pm 0.2035$ | $0.1906 \pm 0.0857$ | $0.8491 \pm 0.0424$ | $0.7466 \pm 0.0168$ | $0.5645 \pm 0.0323$ | $0.4358 \pm 0.0538$ |
| infoGraph+DSVDD | $0.5662 \pm 0.0597$ | $0.7926 \pm 0.0986$ | $0.4062 \pm 0.0978$ | $0.8805 \pm 0.0448$ | $0.6166 \pm 0.2052$ | $0.5312 \pm 0.1545$ | $0.5082 \pm 0.0704$ |
| GLocalKD | $0.4638 \pm 0.0003$ | $0.4330 \pm 0.0016$ | $0.4792 \pm 0.0004$ | $0.3952 \pm 0.2258$ | $0.2965 \pm 0.2641$ | $0.5781 \pm 0.1790$ | $\mathbf{0.7154 \pm 0.0000}$ |
| OCGTL | $0.6504 \pm 0.0433$ | $0.8908 \pm 0.0239$ | $0.4029 \pm 0.0541$ | $0.6570 \pm 0.0210$ | $0.7579 \pm 0.2212$ | $0.2755 \pm 0.0317$ | $0.6915 \pm 0.0207$ |
| DOHSC | $\mathbf{0.9185 \pm 0.0455}$ | $\mathbf{0.9755 \pm 0.0030}$ | $\mathbf{0.8826 \pm 0.0250}$ | $\mathbf{0.8822 \pm 0.0432}$ | $\mathbf{0.8115 \pm 0.0279}$ | $\mathbf{0.6620 \pm 0.0308}$ | $0.5184 \pm 0.0793$ |
| DO2HSC | $\mathbf{0.9390 \pm 0.0025}$ | $\mathbf{0.9836 \pm 0.0115}$ | $\mathbf{0.8835 \pm 0.0118}$ | $\mathbf{0.9089 \pm 0.0609}$ | $\mathbf{0.8250 \pm 0.0790}$ | $\mathbf{0.6867 \pm 0.0226}$ | $\mathbf{0.7351 \pm 0.0159}$ |

Kashima, 2009), OCGIN (Zhao & Akoglu, 2021), infoGraph+Deep SVDD (Sun et al., 2020; Ruff et al., 2018), GLocalKD (Ma et al., 2022) and OCGTL (Qiu et al., 2022).

**Results:** Table 3 shows the comparable results of graph-level anomaly detection. **1)** The proposed methods achieved the best AUC values compared to the other algorithms on all datasets. Both outperform the other state-of-the-art baselines. **2)** DO2HSC is obviously more effective than DOHSC, especially since we observed that there exists a large improvement (exceeding 20%) in Class 1 of ER_MD between DOHSC and DO2HSC. A distance distribution visualization is provided to show their differences in Figure 5. Owing to length limitations, please refer to Appendix H for the remaining results. **3)** The anomaly detection visualization results of DO2HSC displayed in Figure 6 also demonstrate excellent performance. We drew them by setting the projection dimension to 3, and please refer to Appendix I for the results of different perspectives.

### 4.4 More Results and Analysis

We provide the **time and space complexity analysis** in Appendix B. Also, the **ablation study** (including orthogonal projection, mutual information maximization, etc.), **parameter sensitivity** (e.g., different percentile settings), robustness analysis, and more visualization results are shown in Appendices J and I, respectively.

## 5 Conclusion

This paper proposes two novel end-to-end AD methods, DOHSC and DO2HSC, that mitigate the possible shortcomings of hypersphere boundary learning by applying an orthogonal projection for global representation. Furthermore, DO2HSC projects normal data between the interval areas of two co-centered hyperspheres to significantly alleviate the *soap-bubble* issue and the incompactness of a single hypersphere. We also extended DOHSC and DO2HSC to graph-level anomaly detection, which combines the effectiveness of mutual information between the node level and global features to learn graph representation and the power of hypersphere compression. The comprehensive experimental results strongly demonstrate the superiority of the DOHSC and DO2HSC on multifarious datasets. One limitation of this work is that we did not consider cases in which the training data consisted of multiple classes of normal data, which is beyond the scope of this study. Our source code is available at `https://github.com/wownice333/DOHSC-DO2HSC`.

ACKNOWLEDGEMENTS

This work was supported by the National Natural Science Foundation of China under Grant No. 62376236, the General Program JCYJ20210324130208022 of Shenzhen Fundamental Research, the research funding T00120210002 of Shenzhen Research Institute of Big Data, and the funding UDF01001770 of The Chinese University of Hong Kong, Shenzhen.

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

## A  SUPPLEMENTED ALGORITHM PROCEDURES

Here, we present the detailed procedures for DOHSC and DO2HSC in Algorithms 1 and 2, respectively. It begins with a representation learning module and promotes the training data to approximate the center of a hypersphere while adding an orthogonal projection layer. In addition, DO2HSC is recapped in Algorithm 2 and begins with the same representation learning. In contrast, DOHSC utilizes a few epochs to initialize the decision boundaries, after which improved optimization is applied. A graph-level extension is presented in Algorithm 3. The main difference is that graph representation learning with maximization of the mutual information constraint is applied to substitute the common representation learning module. Similarly, the graph-level DO2HSC is the combination of the representation learning part in the graph-level DOHSC and the anomaly detection part in the common DO2HSC.

## B  TIME AND SPACE COMPLEXITY

The models of DOHSC and DO2HSC can be trained by mini-batch optimization. Suppose the batch size is $b$, the maximum width of the hidden layers of the $L$-layer neural network is $w_{\max}$, and the dimension of the input data is $d$, then the time complexities of the proposed methods are at

---

**Algorithm 1** Deep Orthogonal Hypersphere Contraction (DOHSC)

---

**Input:** The input data $\mathbf{X} \in \mathbb{R}^{n \times d}$, dimensions of the latent representation $k$ and orthogonal projection layer $k'$, a trade-off parameter $\lambda$ and the coefficient of regularization term $\mu$, pretraining epoch $\mathcal{T}$, learning rate $\eta$.

**Output:** The anomaly detection scores $\mathbf{s}$.

1: Initialize the auto-encoder network parameters $\mathcal{W} = \{\mathbf{W}_l, \mathbf{b}_l\}_{l=1}^{L}$ and the orthogonal projection layer parameter $\Theta$;
2: **for** $t \to \mathcal{T}$ **do**
3:    **for** each batch **do**
4:       Obtain the latent representation $\mathbf{Z} = f_{\mathcal{W}}^{\text{enc}}(\mathbf{X})$;                        ▷ Pretraining Stage
5:       Update the orthogonal parameter $\Theta$ of orthogonal projection layer by Eq. (3);
6:       Project the latent representation via Eq. (2);
7:       Calculate reconstruction loss via $\frac{1}{n} \sum_{i=1}^{n} \|f_{\mathcal{W}}^{\text{dec}}(\text{Proj}_{\Theta}(f_{\mathcal{W}}^{\text{enc}}(\mathbf{x}_i))) - \mathbf{x}_i\|^2$;
8:       Back-propagate the network, update $\mathcal{W}$ and $\Theta$, respectively;
9:    **end for**
10: **end for**
11: Initialize the center of hypersphere by $\mathbf{c} = \frac{1}{n} \sum_{i=1}^{n} f_{\mathcal{W}}^{\text{enc}}(\mathbf{x}_i)$;
12: **repeat**
13:    **for** each batch **do**
14:       Calculate anomaly detection loss via Optimization (4);                        ▷ Training Stage
15:       Repeat steps 4-6;
16:       Back-propagate the encoder network and update $\{\mathcal{W}\}_{l=1}^{\frac{L}{2}}$ and $\Theta$, respectively;
17:    **end for**
18: **until** convergence
19: Compute decision boundary $r$ by Eq. (5);
20: Calculate the anomaly detection scores $\mathbf{s}$ through Eq. (6);
21: **return** The anomaly detection scores $\mathbf{s}$.

---

most $\mathcal{O}(bd w_{\max} LT)$, where $T$ is the total number of iterations. The space complexities are at most $\mathcal{O}(bd + d w_{\max} + (L-1) w_{\max}^2)$. We see that the complexities are linear with the number of samples, which means the proposed methods are scalable to large datasets. Particularly, for high-dimensional data (very large $d$), we can use small $w_{\max}$ to improve the efficiency.

## C  RELATED PROOF OF BI-HYPERSPHERE LEARNING MOTIVATION

The traditional idea of detecting outliers is to inspect the distribution tails under the ideal assumption that the normal data are Gaussian. Following this assumption, one may argue that an anomalous sample can be distinguished by its large Euclidean distance from the data center ($\ell_2$ norm $\|\mathbf{z} - \mathbf{c}\|$, where $\mathbf{c}$ denotes the centroid). Accordingly, the abnormal dataset is $\{\mathbf{z} : \|\mathbf{z} - \mathbf{c}\| > r\}$ for a decision boundary $r$. However, in high dimensional space, Gaussian distributions look like soap-bubble [3], which means the normal data are more likely to be located in a bi-hypersphere (Vershynin, 2018), such as $\{\mathbf{z} : r_{\min} < \|\mathbf{z} - \mathbf{c}\| < r_{\max}\}$. To better understand this counterintuitive behavior, we generate normal samples $\mathbf{X} \sim \mathcal{N}(\mathbf{0}, \mathbf{I}_d)$, where $d$ is the data dimension in $\{1, 10, 50, 100, 200, 500\}$. As shown in Figure 3 of Section 2.2.1, it indicates that only the univariate Gaussian has a near-zero mode, whereas other high-dimensional Gaussian distributions leave many off-center spaces in the blank. The soap-bubble problem in high-dimensional distributions is well demonstrated in Table 4; the higher the dimension, the greater the quantity of data further away from the center, especially for a 0.01-quantile distance. Thus, we cannot make the sanguine assumption that **all** of the normal data are located within the radius of a hypersphere (i.e., $\{\mathbf{z} : \|\mathbf{z} - \mathbf{c}\| < r\}$). Using Lemma 1 of (Laurent & Massart, 2000), we can prove that Proposition 1, which matches the values in Table 4 that when the dimension is larger, normal data are more likely to lie away from the center.

We also simulated a possible case of outlier detection, in which data were all sampled from a 16-dimensional Gaussian with orthogonal covariance:10,000 normal samples follow $\mathcal{N}(\mathbf{0}, \mathbf{I})$, the first group of 1,000 outliers is from $\mathcal{N}(\mu_1, \frac{1}{10}\mathbf{I})$, the second group of 500 outliers are from $\mathcal{N}(\mu_2, \mathbf{I})$,

---

[3]https://www.inference.vc/high-dimensional-gaussian-distributions-are-soap-bubble/

---

**Algorithm 2** Deep Orthogonal Bi-Hypersphere Compression (DO2HSC)

---

**Input:** The input data $\mathbf{X} \in \mathbb{R}^{n \times d}$, dimensions of the latent representation $k$ and orthogonal projection layer $k'$, a trade-off parameter $\lambda$ and the coefficient of regularization term $\mu$, pretraining epoch $\mathcal{T}_1$, iterations of initializing decision boundaries $\mathcal{T}_2$, learning rate $\eta$.

**Output:** The anomaly detection scores $\mathbf{s}$.

    Initialize the auto-encoder network parameters $\mathcal{W} = \{\mathbf{W}_l, \mathbf{b}_l\}_{l=1}^L$ and the orthogonal projection layer parameter $\Theta$;

 2: **for** $t \to \mathcal{T}_1$ **do**

    **for** each batch **do**

 4:    Repeat steps 4-8 of DOHSC;               ▷ Pretraining Stage

    **end for**

 6: **end for**

    Update the orthogonal parameter $\Theta$ of orthogonal projection layer by Eq. (3);

 8: Obtain the global orthogonal latent representation by Eq. (2);

    Initialize the center of hypersphere by $\mathbf{c} = \frac{1}{n} \sum_{i=1}^n f_{\mathcal{W}}^{\text{enc}}(\mathbf{x}_i)$;

10: **for** $t \to \mathcal{T}_2$ **do**

    Repeat steps 13-17 of DOHSC;             ▷ Pretraining Stage

12: **end for**

    Compute decision boundary $r$ of DOHSC by Eq. (5);

14: Initialize decision boundaries $r_{\max}$ and $r_{\min}$ via Eq. (7);

    **repeat**

16:   **for** each batch **do**

      Obtain the latent representation $\mathbf{Z} = f_{\mathcal{W}}^{\text{enc}}(\mathbf{X})$;      ▷ Training Stage

18:     Update the orthogonal parameter $\Theta$ of orthogonal projection layer by Eq. (3);

      Project the latent representation via Eq. (2);

20:     Calculate the improved total loss via Optimization (8);

      Back-propagate the network, update $\{\mathcal{W}\}_{l=1}^{\frac{L}{2}}$ and $\Theta$, respectively;

22:  **end for**

    **until** convergence

24: Calculate the anomaly detection scores $\mathbf{s}$ through Eq. (9);

    **return** The anomaly detection scores $\mathbf{s}$.

---

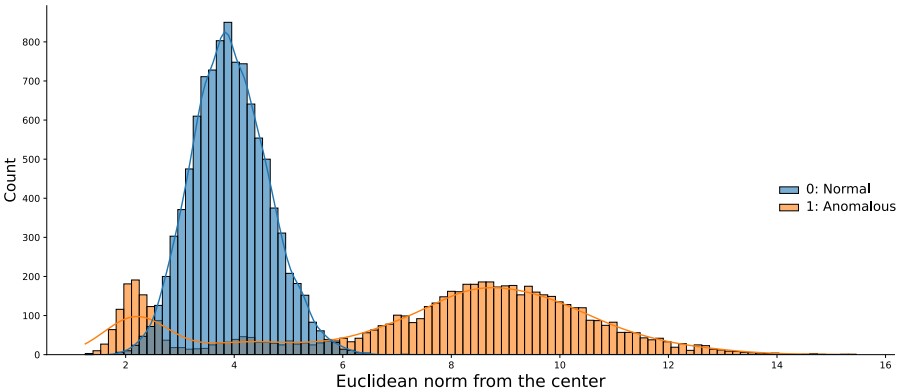

Figure 7: Histogram of distances (Euclidean norm) from the center of normal samples under 16-dimensional Gaussian distributions $\mathcal{N}(\mathbf{0}, \mathbf{I})$. Three groups of anomalous data are also 16-dimensional and respectively sampled from $\mathcal{N}(\mu_1, \frac{1}{10}\mathbf{I})$, $\mathcal{N}(\mu_2, \mathbf{I})$, and $\mathcal{N}(\mu_3, 5\mathbf{I})$, where the population means $\mu_1, \mu_2, \mu_3$ are randomized within $[0, 1]$ for each dimension.

and the last group of 2,000 outliers are from $\mathcal{N}(\mu_3, 5\mathbf{I})$. Figure 7 shows that abnormal data from other distributions (group-1 outliers) could fall a small distance away from the center of the normal samples.

---

**Algorithm 3** Graph-Level Deep Orthogonal Hypersphere Contraction

---

**Input:** The input graph set $\mathbb{G}$, dimensions of GIN hidden layers $k$ and orthogonal projection layer $k'$, a trade-off parameter $\lambda$ and the coefficient of regularization term $\mu$, pretraining epoch $\mathcal{T}$, learning rate $\eta$.

**Output:** The anomaly detection scores $\mathbf{s}$.

    Initialize the network parameters $\Phi$, $\Psi$, $\Upsilon$ and the orthogonal projection layer parameter $\Theta$;

    **for** $t \rightarrow \mathcal{T}$ **do**

3:    **for** each batch $\mathbf{G}$ **do**

        Obtain the global graph representation $\mathbf{H}_{\Phi,\Psi}(G)$;

        Update the orthogonal parameter $\Theta$ of orthogonal projection layer by Eq. (3);

6:        Project the global graph representation via $\tilde{\mathbf{H}}_{\Phi,\Psi,\Theta}(G) = \mathrm{Proj}_\Theta(\mathbf{H}_{\Phi,\Psi}(G))$;

        Calculate $I_{\Phi,\Psi,\Upsilon}\left(\mathbf{h}_{\Phi,\Upsilon}, \tilde{\mathbf{H}}_{\Phi,\Psi}(\mathbf{G})\right)$ via Eq. (13);

        Back-propagate GIN, update $\Phi$, $\Psi$, $\Theta$ and $\Upsilon$, respectively;

9:    **end for**

    **end for**

    Initialize the center of hypersphere by Eq. (14);

12: **repeat**

    **for** each batch $\mathbf{G}$ **do**

        Repeat steps 4-6;

15:    Calculate total loss via Optimization (15);

        Back-propagate GIN and update $\Phi$, $\Psi$, $\Upsilon$ and $\Theta$, respectively;

    **end for**

18: **until** convergence

    Compute decision boundary $r$ by Eq. (5);

    Calculate the anomaly detection scores $\mathbf{s}$ through Eq. (6);

21: **return** The anomaly detection scores $\mathbf{s}$.

---

Table 4: Offcenter distance under multivariate Gaussian at different dimensions and quantiles.

| Quantile (correspond to $r_{\min}$) | dim=1 | dim=10 | dim=50 | dim=100 | dim=200 | dim=500 |
|---|---|---|---|---|---|---|
| 0.01 | 0.0127 | 1.5957 | 5.5035 | 8.3817 | 12.5117 | 20.6978 |
| 0.25 | 0.3115 | 2.5829 | 6.5380 | 9.4908 | 13.6247 | 21.8542 |
| 0.50 | 0.6671 | 3.0504 | 7.0141 | 9.9662 | 14.1054 | 22.3337 |
| 0.75 | 1.1471 | 3.5399 | 7.5032 | 10.4386 | 14.5949 | 22.8200 |
| 0.99 | 2.5921 | 4.8265 | 8.7723 | 11.6049 | 15.7913 | 24.0245 |

## D   Proof for Proposition 2

*Proof.* Since $f$ makes $\mathbf{s}$ obey $\mathcal{N}(\bar{\mathbf{c}}, \mathbf{I}_k)$, according to Proposition 1, we have

$$\mathbb{P}\left[\|\mathbf{s} - \bar{\mathbf{c}}\| \geq \sqrt{k - 2\sqrt{kt}}\right] \geq 1 - e^{-t}.$$

Since $f$ is $\eta$-Lipschitz, we have

$$\|\mathbf{s} - f(\mathbf{c})\| = \|f(\mathbf{z}) - f(\mathbf{c})\| \leq \eta\|\mathbf{z} - \mathbf{c}\|.$$

It follows that

$$\begin{aligned}\|\mathbf{z} - \mathbf{c}\| &\geq \eta^{-1}\|\mathbf{s} - \bar{\mathbf{c}} + \bar{\mathbf{c}} - f(\mathbf{c})\| \\ &\geq \eta^{-1}\left(\|\mathbf{s} - \bar{\mathbf{c}}\| - \|\bar{\mathbf{c}} - f(\mathbf{c})\|\right) \\ &\geq \eta^{-1}\left(\|\mathbf{s} - \bar{\mathbf{c}}\| - \epsilon\right).\end{aligned}$$

Now we have

$$\mathbb{P}\left[\|\mathbf{z} - \mathbf{c}\| \geq \eta^{-1}\left(\sqrt{k - 2\sqrt{kt}} - \epsilon\right)\right] \geq 1 - e^{-t}.$$

This finished the proof. $\qquad\square$

## E   PROOF FOR PROPOSITION 3

*Proof.* The volume of a hyperball of radius $r$ in $k$-dimension space is $V_k(r) = \frac{\pi^{k/2}}{\Gamma\left(\frac{k}{2}+1\right)}r^k$, where $\Gamma$ is Euler's gamma function. Then, the volume of the hypersphere given by DOHSC is $V_{\max} = \frac{\pi^{k/2}}{\Gamma\left(\frac{k}{2}+1\right)}r_{\max}^k$ and the volume of the smaller hypersphere given by DO2HSC is $V_{\min} = \frac{\pi^{k/2}}{\Gamma\left(\frac{k}{2}+1\right)}r_{\min}^k$. Then, the volume of the decision region given by DO2HSC is $V_{\max} - V_{\min}$. The density of the decision region is defined as the number of normal data in unit volume. Therefore, the ratio between the densities of DO2HSC and DOHSC can be computed as

$$\rho = \frac{n/(V_{\max} - V_{\min})}{n/V_{\max}} = \frac{\frac{\pi^{k/2}}{\Gamma\left(\frac{k}{2}+1\right)}r_{\max}^k}{\frac{\pi^{k/2}}{\Gamma\left(\frac{k}{2}+1\right)}r_{\max}^k - \frac{\pi^{k/2}}{\Gamma\left(\frac{k}{2}+1\right)}r_{\min}^k} = \frac{1}{1 - \left(\frac{r_{\min}}{r_{\max}}\right)^k}. \tag{17}$$

This finished the proof. □

In the case of $r_{\min} = r_{\max}$, the volume of DO2HSC is close to an infinitely thin shell, essentially transforming into the surface of a hypersphere. In this scenario, the data density of DO2HSC is significantly higher compared with that of DOHSC. However, it is important to note that this situation is quite rare, particularly in high-dimensional space.

## F   EXPERIMENT CONFIGURATION

In this section, experimental settings are presented for reproduction. First, each graph dataset was divided into two parts: the training and testing sets. We randomly sampled 80 percent of the normal graph as the training set and the remaining normal graph, together with the randomly sampled abnormal data in a one-to-one ratio to form the testing set. Regarding the image and tabular datasets, the data splits are already provided in the paper. The detailed statistical information of all tested datasets is given in Tables 5 and 6.

Table 5: Description for non-graph datasets.

| Dataset Name | Type | # Instances | # Dimension |
|---|---|---|---|
| Thyroid | Tabular | 3772 | 6 |
| Arrhythmia | Tabular | 452 | 274 |
| Fashion-MNIST | Image | 70000 | $28 \times 28$ |
| CIFAR-10 | Image | 60000 | $32 \times 32 \times 3$ |

Table 6: Description for six graph datasets.

| Datasets | # Graphs | Avg. # Nodes | Avg. # Edges | # Classes | # Graph Labels |
|---|---|---|---|---|---|
| COLLAB | 5000 | 74.49 | 2457.78 | 3 | 2600 / 775 / 1625 |
| COX2 | 467 | 42.43 | 44.54 | 2 | 365 / 102 |
| ER_MD | 446 | 21.33 | 234.85 | 2 | 265 / 181 |
| MUTAG | 188 | 17.93 | 19.79 | 2 | 63 / 125 |
| DD | 1178 | 284.32 | 715.66 | 2 | 691 / 487 |
| IMDB-Binary | 1000 | 19.77 | 96.53 | 2 | 500 / 500 |

In the image and tabular experiments, our backbone was consistent with the Deep SVDD, and the preprocessing conformed to the same splits as DROCC. All results originated from the corresponding papers or were reproduced according to the official code. Regarding our DOHSC model, we set 10 epochs in the pretraining stage to initialize the center of the decision boundary and then train the model in 200 epochs. The percentile $\nu$ of $r$ was selected from

$\{0.001, 0.003, 0.005, 0.008, 0.01, 0.03, 0.1, 0.3\}$. The improved method DO2HSC also sets a 10-epoch pretraining stage and trains DOHSC for 50 epochs to initialize a suitable center and decision boundaries $r_{\max}$ and $r_{\min}$, where the percentile $\nu$ of $r_{\max}$ is the same as DOHSC. The main training epoch was set to 200.

In the graph experiment, we adopted the classical AD method, One-Class SVM (OCSVM) (Schölkopf et al., 2001), to compare graph-kernel baselines and used 10-fold cross-validation to make a fair comparison. All graph kernels extract a kernel matrix via GraKel (Siglidis et al., 2020) and apply the OCSVM in scikit-learn (Pedregosa et al., 2011). Specifically, we selected Floyd Warshall as the SP kernel's algorithm and set lambda as 0.01 for the RW kernel. The WL kernel algorithm is sensitive to the number of iterations; therefore, we tested four different iterations $\{2, 5, 8, 10\}$ and reported the best result for each experiment. The outputs were normalized for all the graph kernels. For infoGraph+Deep SVDD, the first stage runs for 20 epochs, and the second stage pretrains for 50 epochs and trains for 100 epochs. In OCGIN, GLocalKD, and OCGTL, the default or reported parameter settings were adopted to reproduce the experimental results.

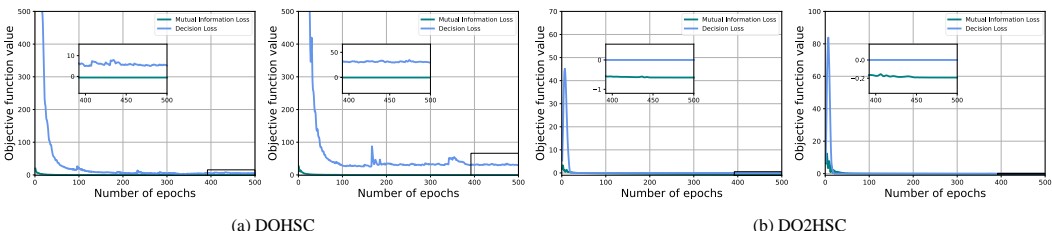

(a) DOHSC                 (b) DO2HSC

Figure 8: Convergence curves of the proposed models on the MUTAG dataset.

For the graph-level DOHSC, we first set one epoch in the pre-training stage to initialize the center of the decision boundary and then train the model in 500 epochs. The convergence curves are shown in Figure 8 to indicate that the final optimized results were adopted. The improved method DO2HSC is also set as a 1-epoch pre-training stage and trains DOHSC for five epochs, where the percentile $\nu$ of $r_{\max}$ is selected as 0.01. After initialization, the model was trained for 500 epochs. For both proposed approaches, the trade-off factor $\lambda$ was set to 10 to ensure decision loss as the main optimization objective. The dimensions of the GIN hidden and orthogonal projection layers were fixed at 16 and 8, respectively. For the backbone network, a 4-layer GIN and a 3-layer fully connected neural network were adopted.

Finally, the averages and standard deviations of the Area Under the ROC curve (AUC) and F1-score were used to support the comparable experiments by repeating each algorithm ten times. A higher metric value indicates better performance.

## G    SUPPLEMENTED RESULTS ON FASHION-MNIST DATASET

The complete experimental results for the Fashion-MNIST image dataset are given in Table 7. A detailed standard deviation can demonstrate fluctuations in performance. The proposed methods are relatively stable, especially for DOHSC.

## H    SUPPLEMENTARY RESULTS OF GRAPH-LEVEL ANOMALY DETECTION

Here, we give the results of retained 3 graph datasets (COX2, DD, and IMDB-Binary) for graph-level extension in Table 8. The proposed two models are superior on all datasets and behave much more effectively compared with other SOTAs, which also supports our motivations for graph-level anomaly detections.

## I    SUPPLEMENTED VISUALIZATION

This section presents the related supplemental visualization results of the anomaly detection task. Figure 9 shows the distance distributions of the two-stage method, proposed model DOHSC, and

Table 7: Average AUCs in one-class anomaly detection on Fashion-MNIST. (The best two results are marked in **bold**.)

| Normal Class | Deep SVDD (Ruff et al., 2018) | DROCC (Goyal et al., 2020) | DOHSC | DO2HSC |
|---|---|---|---|---|
| T-shirt | $0.8263 \pm 0.0342$ | $0.8931 \pm 0.0072$ | $\mathbf{0.9153 \pm 0.0082}$ | $\mathbf{0.9196 \pm 0.0064}$ |
| Trouser | $0.9632 \pm 0.0072$ | $\mathbf{0.9835 \pm 0.0054}$ | $0.9817 \pm 0.0060$ | $\mathbf{0.9839 \pm 0.0020}$ |
| Pullover | $0.7885 \pm 0.0398$ | $\mathbf{0.8656 \pm 0.0140}$ | $0.8007 \pm 0.0204$ | $\mathbf{0.8768 \pm 0.0122}$ |
| Dress | $0.8607 \pm 0.0124$ | $0.8776 \pm 0.0269$ | $\mathbf{0.9178 \pm 0.0230}$ | $\mathbf{0.9171 \pm 0.0084}$ |
| Coat | $0.8417 \pm 0.0366$ | $0.8453 \pm 0.0143$ | $\mathbf{0.8805 \pm 0.0258}$ | $\mathbf{0.9038 \pm 0.0140}$ |
| Sandal | $0.8902 \pm 0.0281$ | $\mathbf{0.9336 \pm 0.0123}$ | $0.8932 \pm 0.0287$ | $\mathbf{0.9308 \pm 0.0070}$ |
| Shirt | $0.7507 \pm 0.0158$ | $0.7789 \pm 0.0188$ | $\mathbf{0.8177 \pm 0.0124}$ | $\mathbf{0.8022 \pm 0.0045}$ |
| Sneaker | $0.9676 \pm 0.0062$ | $0.9624 \pm 0.0059$ | $\mathbf{0.9678 \pm 0.0050}$ | $\mathbf{0.9677 \pm 0.0075}$ |
| Bag | $0.9039 \pm 0.0355$ | $0.7797 \pm 0.0749$ | $\mathbf{0.9122 \pm 0.0258}$ | $\mathbf{0.9090 \pm 0.0105}$ |
| Ankle Boot | $0.9488 \pm 0.0207$ | $0.9589 \pm 0.0207$ | $\mathbf{0.9756 \pm 0.0127}$ | $\mathbf{0.9785 \pm 0.0038}$ |

Table 8: Average AUCs with standard deviation (10 trials) of different graph-level anomaly detection algorithms. 'DSVDD' stands for 'Deep SVDD'. We assess models by regarding every data class as normal data, respectively. The best two results are highlighted in **bold** and '−' means out of memory.

| | COX2 | | DD | | IMDB-Binary | |
|---|---|---|---|---|---|---|
| | 0 | 1 | 0 | 1 | 0 | 1 |
| SP+OCSVM | $0.5408 \pm 0.0000$ | $0.5760 \pm 0.0000$ | $0.6856 \pm 0.0000$ | $0.4474 \pm 0.0000$ | $0.4592 \pm 0.0000$ | $0.4716 \pm 0.0000$ |
| WL+OCSVM | $0.5990 \pm 0.0000$ | $0.5057 \pm 0.0000$ | $0.7397 \pm 0.0000$ | $0.4946 \pm 0.0000$ | $0.5157 \pm 0.0000$ | $0.4607 \pm 0.0000$ |
| NH+OCSVM | $0.4841 \pm 0.0000$ | $0.4717 \pm 0.0000$ | $\mathbf{0.7424 \pm 0.0000}$ | $0.3684 \pm 0.0000$ | $0.5321 \pm 0.0000$ | $0.4652 \pm 0.0000$ |
| RW+OCSVM | $0.5243 \pm 0.0000$ | $0.6553 \pm 0.0000$ | − | − | $0.4951 \pm 0.0000$ | $0.5311 \pm 0.0000$ |
| OCGIN | $0.5964 \pm 0.0578$ | $0.5683 \pm 0.0768$ | $0.6659 \pm 0.0444$ | $0.6003 \pm 0.0534$ | $0.4571 \pm 0.1879$ | $0.3736 \pm 0.0816$ |
| infoGraph+DSVDD | $0.4825 \pm 0.0624$ | $0.5029 \pm 0.0700$ | $0.3942 \pm 0.0436$ | $0.6484 \pm 0.0236$ | $0.6353 \pm 0.0277$ | $0.5836 \pm 0.0995$ |
| GLocalKD | $0.3861 \pm 0.0131$ | $0.3143 \pm 0.0383$ | $0.1952 \pm 0.0000$ | $0.2203 \pm 0.0001$ | $0.5383 \pm 0.0124$ | $0.4812 \pm 0.0101$ |
| OCGTL | $0.5541 \pm 0.0320$ | $0.4862 \pm 0.0224$ | $0.6990 \pm 0.0260$ | $0.6767 \pm 0.0280$ | $0.6510 \pm 0.0180$ | $0.6412 \pm 0.0127$ |
| DOHSC | $\mathbf{0.6263 \pm 0.0333}$ | $\mathbf{0.6805 \pm 0.0168}$ | $0.7083 \pm 0.0188$ | $\mathbf{0.7579 \pm 0.0154}$ | $\mathbf{0.7160 \pm 0.0600}$ | $\mathbf{0.7705 \pm 0.0045}$ |
| DO2HSC | $\mathbf{0.6329 \pm 0.0292}$ | $\mathbf{0.6923 \pm 0.0433}$ | $\mathbf{0.7320 \pm 0.0194}$ | $\mathbf{0.7651 \pm 0.0317}$ | $\mathbf{0.7547 \pm 0.0390}$ | $\mathbf{0.7737 \pm 0.0503}$ |

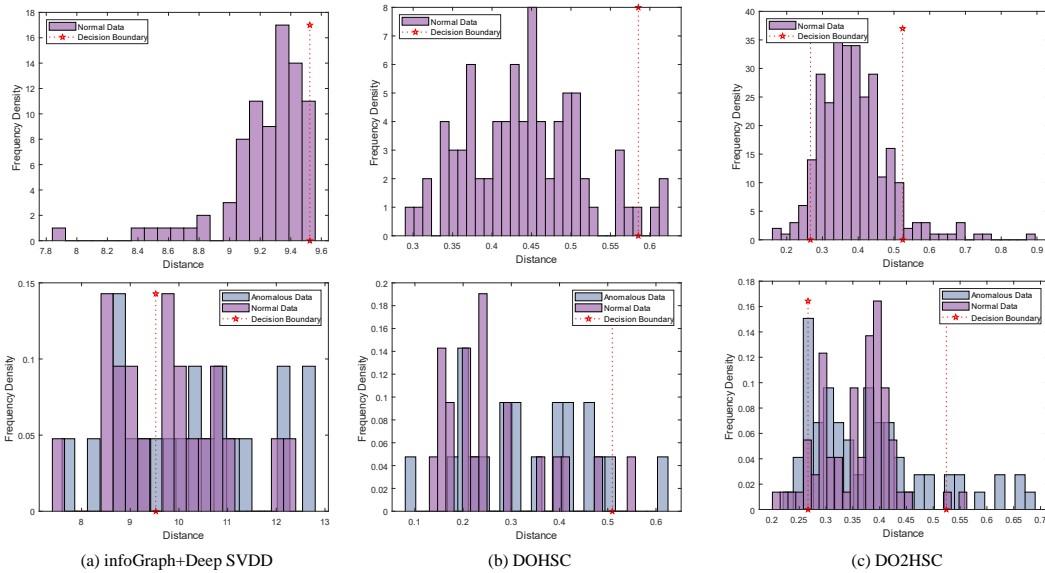

(a) infoGraph+Deep SVDD    (b) DOHSC    (c) DO2HSC

Figure 9: Distance distributions were obtained by infoGraph+Deep SVDD, the proposed model, and the improved proposed model on COX2. The first row represents the distance distribution of the training samples in relation to the decision boundary. The last row indicates the distance distribution of the test data with respect to the decision boundary.

improved DO2HSC. Here, *distance* is defined as the distance between each sample and the center of the decision hypersphere. Distance distribution denotes the sample proportion at this distance interval relative to the corresponding total samples. It can be intuitively observed that most of the distances of the instances were close to the decision boundary because of the fixed learned representation. As mentioned earlier, the jointly trained algorithm mitigated this situation, and the obtained representation caused many instances to have smaller distances from the center of the sphere. Moreover, as mentioned in Section 2.2, anomalous data may occur in regions with less training data, particularly in the region close to the center, which is also confirmed by (a) and (b) of Figure 9. In contrast, DO2HSC effectively shrinks the decision area, and we find that the number of outliers is obviously reduced owing to a more compact distribution of the training data.

The 3D visualization results of the training and testing stages are also presented the difference between them in Figures 10 and 11.

To further support the aforementioned statements, as shown in Figure 12, the anomalous samples are located in the decision region and are closer to the center than other normal samples. On the contrary, the result of DO2HSC effectively prevents this phenomenon.

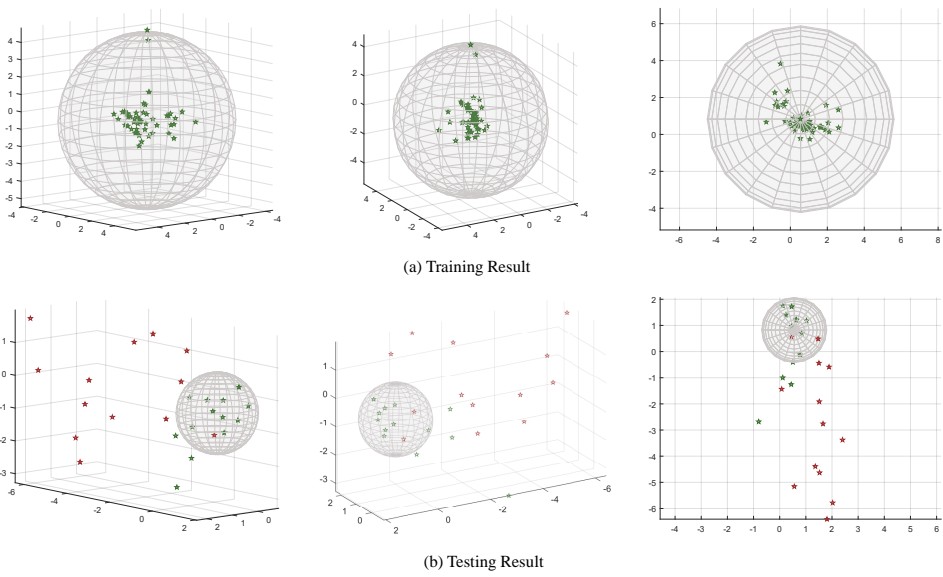

(a) Training Result

(b) Testing Result

Figure 10: Visualization results of the DOHSC with MUTAG in different perspectives.

## J  PARAMETER SENSITIVITY AND ROBUSTNESS

To confirm the stability of our models, we analyzed the parameter sensitivity and robustness of DOHSC and DO2HSC, respectively. Consider that the projection dimension varies in {4, 8, 16, 32, 64, 128}, whereas the hidden layer dimension of the GIN module ranges from 4 to 128. In Figure 13, the DO2HSC model has less volatile performance than DOHSC, especially when the training dataset is sampled from COX2 class 0, as shown in Subfigure (d). Noticeably, a higher dimension of the GIN hidden layer usually displays a better AUC result because the quality of the learned graph representations improves when the embedding space is sufficiently large.

In addition, we assessed different aspects of model robustness. More specifically, the AUC results about two "ratios" are displayed: 1) Different sampling ratios for the training set; 2) Different ratios of noise disturbance for the learned representation. In Subfigures (c) and (f), the purple bars regard normal data as class 0, whereas green bars treat normal data as class 1. Note that most AUC results are elevated along with a higher ratio of authentic data in the training stage, demonstrating the potential of our models in the unsupervised setting. On the other hand, when more noise is blended into the training dataset, the AUC performances of the yellow line and blue line always remain stable at a high level. This outcome verifies the robustness of our model in response to alien data.

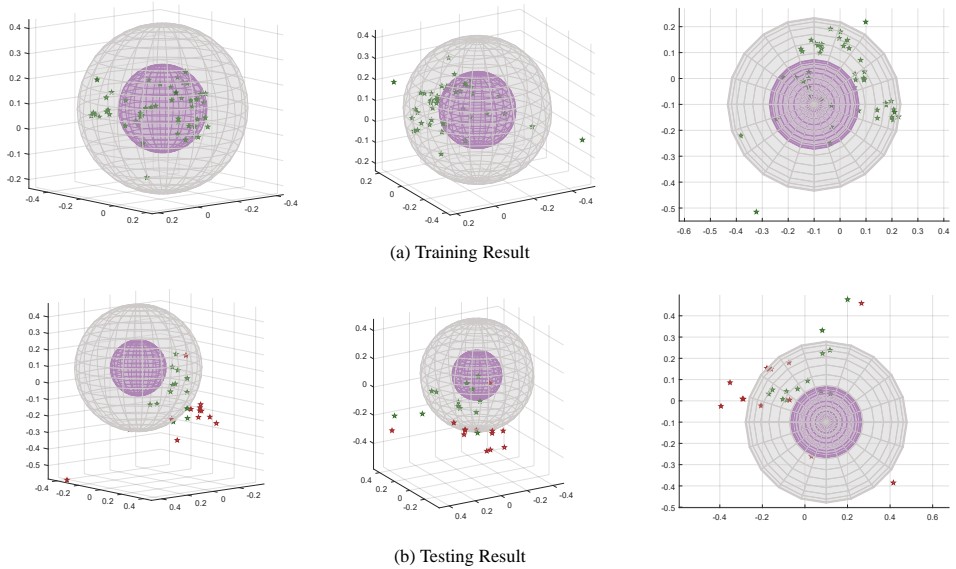

(a) Training Result

(b) Testing Result

Figure 11: Visualization results of the DO2HSC with MUTAG in different perspectives.

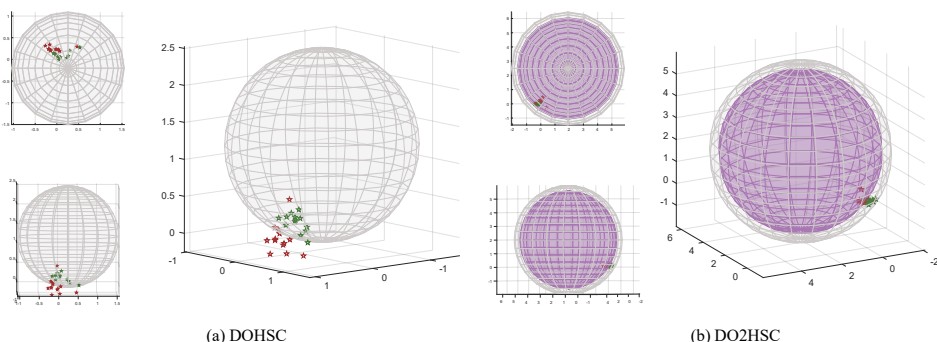

(a) DOHSC

(b) DO2HSC

Figure 12: Anomaly detection comparison between DOHSC and DO2HSC on MUTAG.

The percentile parameter sensitivity is presented in this section. It is worth mentioning that we tested DOHSC with varying percentiles in $\{0.01, 0.1, ..., 0.8\}$ and tested DO2HSC only in $\{0.01, 0.05, 0.1\}$ because the two radii of DO2HSC are obtained by the percentiles $\nu$ and $1 - \nu$. The two radii are equal when $\nu = 0.5$ and the interval between the two co-centered hyperspheres disappears. From the table, the performance would decrease when a larger percentile is set obviously. For example, on the MUTAG dataset, setting the percentile as 0.01 is more beneficial for DOHSC than setting it as 0.8, and setting the percentile as 0.01 is better than setting it as 0.1 for DO2HSC due to the change of the interval area.

Table 9: Parameter sensitivity of **DOHSC** with different percentiles (all normal data is set to Class 0.)

| Dataset | Percentile | | | | |
|---|---|---|---|---|---|
| | 0.005 | 0.01 | 0.1 | 0.5 | 0.8 |
| COX2 | 0.5446 (0.0854) | **0.6263 (0.0333)** | 0.6022 (0.0789) | 0.5232 (0.0494) | 0.5523 (0.0572) |
| ER_MD | 0.6265 (0.1442) | 0.6620 (0.0308) | **0.7497 (0.0411)** | 0.6265 (0.1442) | 0.5141 (0.0398) |
| MUTAG | 0.8185 (0.0543) | **0.8822 (0.0432)** | 0.8540 (0.0694) | 0.7790 (0.0912) | 0.8675 (0.1287) |
| DD | 0.6349 (0.0380) | **0.7083 (0.0188)** | 0.6597 (0.0270) | 0.6545 (0.0268) | 0.6327 (0.0206) |
| IMDB-Binary | **0.7232 (0.0314)** | 0.7160 (0.0600) | 0.7217 (0.0418) | 0.7073 (0.0274) | 0.6773 (0.0566) |

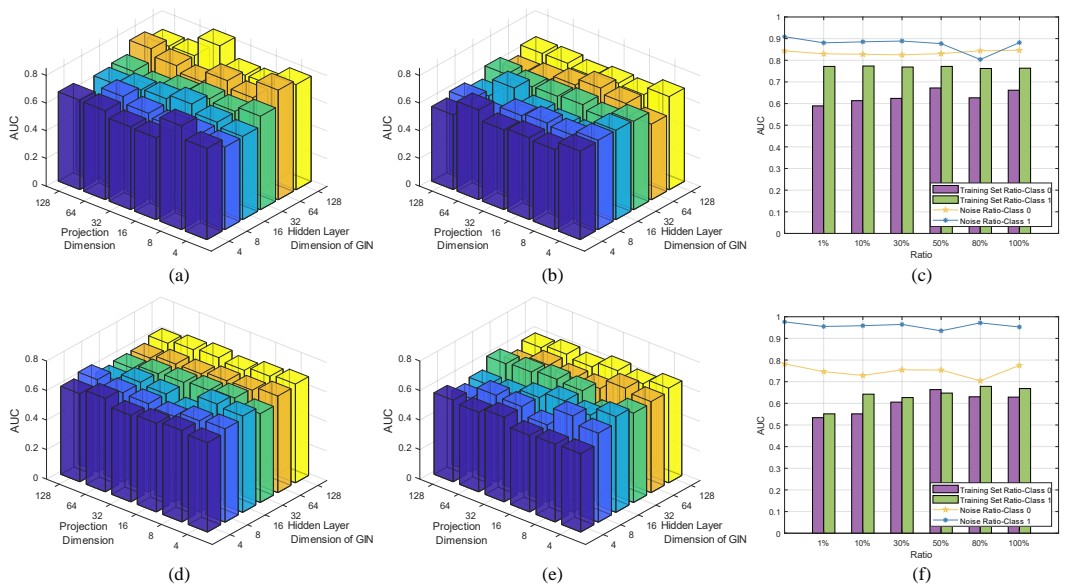

Figure 13: Parameter sensitivity and robustness of the proposed models. (a)-(b) Parameter sensitivity of DOHSC with different hidden layer dimensions of GIN and projection dimensions on COX2 with Class 0 and Class 1, respectively. (d)-(e) Parameter sensitivity of DO2HSC with the same settings. (c) and (f) shows the performance impacts with different ratios of the training set on the IMDB-Binary dataset and added noise disturbances on the MUTAG dataset while training DOHSC and DO2HSC, respectively.

Table 10: Parameter sensitivity of **DO2HSC** with different percentiles (all normal data is set to Class 0.)

| Dataset | Percentile | | | |
|---|---|---|---|---|
| | 0.005 | 0.01 | 0.05 | 0.1 |
| COX2 | 0.5810 (0.0354) | **0.6329 (0.0292)** | 0.6149 (0.0187) | 0.5830 (0.0713) |
| ER_MD | 0.6136 (0.0769) | 0.6226 (0.0890) | **0.6867 (0.0226)** | 0.6331 (0.1748) |
| MUTAG | 0.7278 (0.0478) | **0.9089 (0.0609)** | 0.8041 (0.1006) | 0.6769 (0.1207) |
| DD | 0.7103 (0.0098) | **0.7320 (0.0194)** | 0.6909 (0.0208) | 0.6765 (0.0286) |
| IMDB-Binary | **0.6590 (0.0287)** | 0.6406 (0.0642) | 0.5348 (0.0486) | 0.5701 (0.0740) |

## K  SUPPLEMENTED RESULTS OF ABLATION STUDY

First, an ablation study of whether orthogonal projection requires standardization was conducted. More precisely, we pursue orthogonal features, that is, finding a projection matrix for orthogonal latent representation (with standardization) instead of computing the projection onto the column or row space of the projection matrix (non-standardization), although they are closely related to each other. This is equivalent to performing PCA and using standardized principal components. Therefore, we compared the DOHSC with and without standardization. From Table 11, **1)** it is observed that the performance of DOHSC without standardization is acceptable, and most of its results are better than those of the two-stage baseline, i.e., infoGraph+Deep SVDD. This verifies the superiority of the end-to-end method over two-stage baselines. **2)** The standardized model results outperform the non-standardized model results in all cases. **3)** DO2HSC surpasses DOHSC no matter with/without the orthogonal projection layer.

Besides, the ablation study using the mutual information maximization loss is shown in Table 12. It can be intuitively concluded that mutual information loss does not always have a positive effect on all data. This also indicates that the designed anomaly detection optimization method and orthogonal projection are effective, instead of entirely, owing to the loss of mutual information.

Table 11: Comparison of the orthogonal projection layer with or w/o standardization. 'DSVDD' stands for 'Deep SVDD'. 'Non-Std stands for 'Non-Standardization'.

| | Class | infoGraph+DSVDD | DOHSC (Non-Std) | DOHSC | DO2HSC (Non-Std) | DO2HSC |
|---|---|---|---|---|---|---|
| MUTAG | 0 | $0.8805 \pm 0.0448$ | $0.8521 \pm 0.0650$ | $\mathbf{0.8822 \pm 0.0432}$ | $0.9024 \pm 0.0207$ | $\mathbf{0.9089 \pm 0.0609}$ |
| | 1 | $0.6166 \pm 0.2052$ | $0.6918 \pm 0.1467$ | $\mathbf{0.8115 \pm 0.0279}$ | $0.7624 \pm 0.0248$ | $\mathbf{0.8250 \pm 0.0790}$ |
| COX2 | 0 | $0.4825 \pm 0.0624$ | $0.5800 \pm 0.0473$ | $\mathbf{0.6263 \pm 0.0333}$ | $0.6127 \pm 0.0191$ | $\mathbf{0.6329 \pm 0.0292}$ |
| | 1 | $0.5029 \pm 0.0700$ | $0.5029 \pm 0.0697$ | $\mathbf{0.6805 \pm 0.0168}$ | $0.6303 \pm 0.0276$ | $\mathbf{0.6923 \pm 0.0433}$ |
| ER_MD | 0 | $0.5312 \pm 0.1545$ | $0.4881 \pm 0.0626$ | $\mathbf{0.6620 \pm 0.0308}$ | $0.6148 \pm 0.0484$ | $\mathbf{0.6867 \pm 0.0226}$ |
| | 1 | $0.5082 \pm 0.0704$ | $0.5140 \pm 0.0356$ | $\mathbf{0.5184 \pm 0.0793}$ | $0.7043 \pm 0.0011$ | $\mathbf{0.7351 \pm 0.0159}$ |
| DD | 0 | $0.3942 \pm 0.0436$ | $0.4029 \pm 0.0354$ | $\mathbf{0.7083 \pm 0.0188}$ | $0.7308 \pm 0.0015$ | $\mathbf{0.7320 \pm 0.0194}$ |
| | 1 | $0.6484 \pm 0.0236$ | $0.6903 \pm 0.0215$ | $\mathbf{0.7579 \pm 0.0154}$ | $0.7000 \pm 0.0165$ | $\mathbf{0.7651 \pm 0.0317}$ |
| IMDB-Binary | 0 | $0.6353 \pm 0.0277$ | $0.5149 \pm 0.0655$ | $\mathbf{0.6609 \pm 0.0033}$ | $0.6387 \pm 0.0578$ | $\mathbf{0.7547 \pm 0.0390}$ |
| | 1 | $0.5836 \pm 0.0995$ | $0.6505 \pm 0.0585$ | $\mathbf{0.7705 \pm 0.0045}$ | $0.7032 \pm 0.0328$ | $\mathbf{0.7737 \pm 0.0503}$ |
| COLLAB | 0 | $0.5662 \pm 0.0597$ | $0.6067 \pm 0.1007$ | $\mathbf{0.9185 \pm 0.0455}$ | $0.7089 \pm 0.0335$ | $\mathbf{0.9390 \pm 0.0025}$ |
| | 1 | $0.7926 \pm 0.0986$ | $0.8958 \pm 0.0141$ | $\mathbf{0.9755 \pm 0.0030}$ | $0.9033 \pm 0.0089$ | $\mathbf{0.9836 \pm 0.0115}$ |
| | 2 | $0.4062 \pm 0.0978$ | $0.4912 \pm 0.2000$ | $\mathbf{0.8826 \pm 0.0250}$ | $0.7158 \pm 0.1059$ | $\mathbf{0.8835 \pm 0.0118}$ |

Table 12: Comparison of the loss supervision with or w/o **m**utual **i**nformation **l**oss (**MIL**).

| | Class | DOHSC (Non-**MIL**) | DOHSC | DO2HSC (Non-**MIL**) | DO2HSC |
|---|---|---|---|---|---|
| MUTAG | 0 | $\mathbf{0.9456 \pm 0.0189}$ | $0.8822 \pm 0.0432$ | $0.8308 \pm 0.0548$ | $\mathbf{0.9089 \pm 0.0609}$ |
| | 1 | $0.7597 \pm 0.0802$ | $\mathbf{0.8115 \pm 0.0279}$ | $0.7915 \pm 0.0274$ | $\mathbf{0.8250 \pm 0.0790}$ |
| COX2 | 0 | $\mathbf{0.6349 \pm 0.0466}$ | $0.6263 \pm 0.0333$ | $0.6143 \pm 0.0302$ | $\mathbf{0.6329 \pm 0.0292}$ |
| | 1 | $0.6231 \pm 0.0501$ | $\mathbf{0.6805 \pm 0.0168}$ | $0.6576 \pm 0.1830$ | $\mathbf{0.6923 \pm 0.0433}$ |
| ER_MD | 0 | $0.5837 \pm 0.0778$ | $\mathbf{0.6620 \pm 0.0308}$ | $0.5836 \pm 0.0909$ | $\mathbf{0.6867 \pm 0.0226}$ |
| | 1 | $\mathbf{0.6465 \pm 0.0600}$ | $0.5184 \pm 0.0793$ | $\mathbf{0.7424 \pm 0.0385}$ | $0.7351 \pm 0.0159$ |
| DD | 0 | $0.4738 \pm 0.0412$ | $\mathbf{0.7083 \pm 0.0188}$ | $0.6882 \pm 0.0221$ | $\mathbf{0.7320 \pm 0.0194}$ |
| | 1 | $0.7197 \pm 0.0185$ | $\mathbf{0.7579 \pm 0.0154}$ | $0.7376 \pm 0.0244$ | $\mathbf{0.7651 \pm 0.0317}$ |
| IMDB-Binary | 0 | $0.5666 \pm 0.0810$ | $\mathbf{0.6609 \pm 0.0033}$ | $0.6303 \pm 0.0538$ | $\mathbf{0.7547 \pm 0.0390}$ |
| | 1 | $0.6827 \pm 0.0239$ | $\mathbf{0.7705 \pm 0.0045}$ | $0.6810 \pm 0.0276$ | $\mathbf{0.7737 \pm 0.0503}$ |
| COLLAB | 0 | $\mathbf{0.9330 \pm 0.0539}$ | $0.9185 \pm 0.0455$ | $0.5415 \pm 0.0182$ | $\mathbf{0.9390 \pm 0.0025}$ |
| | 1 | $0.9744 \pm 0.0017$ | $\mathbf{0.9755 \pm 0.0030}$ | $0.9293 \pm 0.0023$ | $\mathbf{0.9836 \pm 0.0115}$ |
| | 2 | $0.8275 \pm 0.0765$ | $\mathbf{0.8826 \pm 0.0250}$ | $0.8452 \pm 0.0243$ | $\mathbf{0.8835 \pm 0.0118}$ |

To demonstrate the effectiveness of the orthogonal projection layer (OPL), we conducted ablation studies and compared the comparison of 3-dimensional results produced with and without the OPL. For each model trained on a particular dataset class, we show the result without OPL on the left side, whereas the result with OPL is displayed on the right. As Figure 14 illustrates, the OPL drastically improves the distribution of the embeddings to be more spherical rather than elliptical. Similarly, with the help of the OPL, the other embeddings exhibited a more compact and rounded layout.

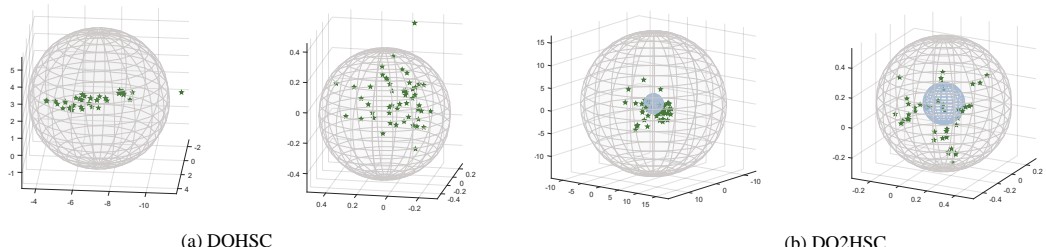

(a) DOHSC      (b) DO2HSC

Figure 14: Visualizations on the MUTAG dataset Class 0 (left: without OPL; right: with OPL).

## L   IMBALANCED EXPERIMENTAL RESULTS

We also give the experiment on graph-level datasets with an imbalanced setting of the ratio between anomalies and normal graphs in the experimental datasets. Please refer to Table 13, which showcases

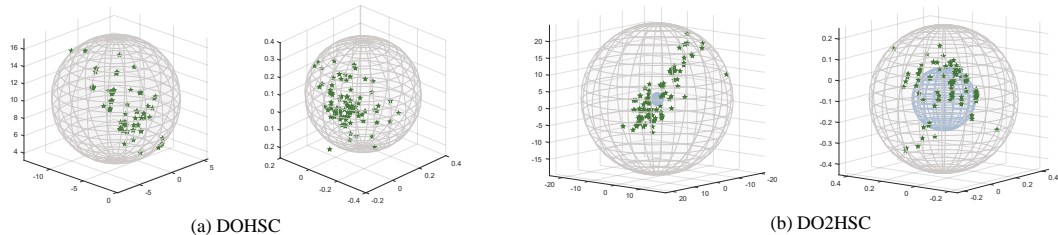

(a) DOHSC                              (b) DO2HSC

Figure 15: Visualizations on the MUTAG dataset Class 1 (left: without OPL; right: with OPL).

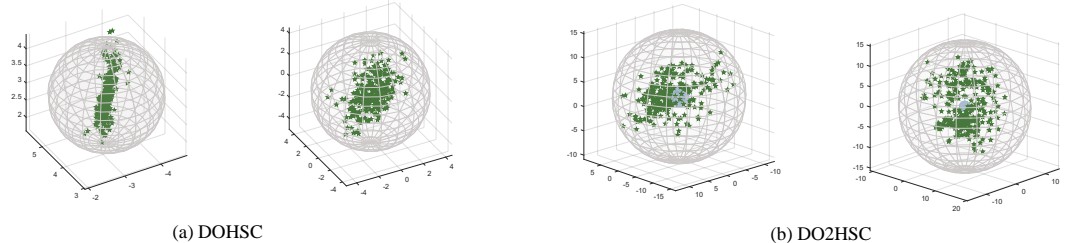

(a) DOHSC                              (b) DO2HSC

Figure 16: Visualizations on the COX2 dataset Class 0 (left: without OPL; right: with OPL).

our results on imbalanced datasets. The experimental results illustrate that the proposed methods overcome the imbalanced problem better than other algorithms in general. But DO2HSC has more advantageous results.

# M  RELATED WORK

## M.1  SOME SOTA ANOMALY DETECTION METHODS

In this section, we first provide an overview of some SOTA anomaly detection methods. The method proposed by Perera et al. (2019) involves adversarial training of an auto-encoder and a discriminator, while compelling the latent representation by one class of data. Goyal et al. (2020) judged the anomalous data according to the assumption that the normal instances generally lie on a low-dimensional locally linear manifold, and regarded the process of finding the decision boundary in the embedding space as an adversarial manner. Hu et al. (2020) combined holistic regularization with a 2-norm instance-level normalization, thus further proposing an effective one-class learning method. Cai & Fan (2022) proposed a perturbation learning based anomaly detection method, which generates the negative samples containing the smallest noise as much as possible, to train a detection classifier. The assumption is that, if this classifier can discriminate this type of negative samples and normal data, it should have the ability to distinguish more different anomalous data. All aforementioned algorithms are compared in our experimental section to support the effectiveness of the proposed improvements.

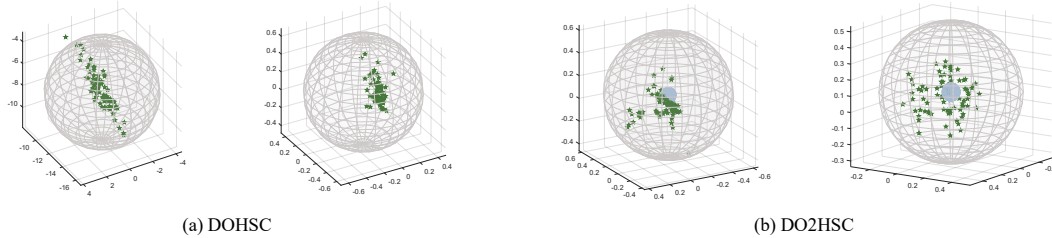

(a) DOHSC                              (b) DO2HSC

Figure 17: Visualizations on the COX2 dataset Class 1 (left: without OPL; right: with OPL).

Table 13: Average AUCs with standard deviation (10 trials) of imbalanced experiments (the ratio of normal data to abnormal data is 10:1). 'DSVDD' stands for 'Deep SVDD'. The best two results are highlighted in **bold** and '−' means out of memory.

| | COX2 | | ER_MD | | MUTAG | |
|---|---|---|---|---|---|---|
| | 0 | 1 | 0 | 1 | 0 | 1 |
| SP+OCSVM | $0.4854 \pm 0.0000$ | $0.7874 \pm 0.0000$ | $0.2814 \pm 0.0000$ | $0.0764 \pm 0.0000$ | $0.2917 \pm 0.0000$ | $0.0266 \pm 0.0000$ |
| WL+OCSVM | $0.4127 \pm 0.0000$ | $0.8125 \pm 0.0000$ | $0.5142 \pm 0.0000$ | $0.1909 \pm 0.0000$ | $0.7083 \pm 0.0000$ | $0.0399 \pm 0.0000$ |
| NH+OCSVM | $0.3818 \pm 0.0385$ | $0.4875 \pm 0.0000$ | $0.5774 \pm 0.0273$ | $0.3215 \pm 0.0274$ | $0.1910 \pm 0.0000$ | $0.0573 \pm 0.0833$ |
| RW+OCSVM | $-$ | $-$ | $0.5220 \pm 0.0000$ | $0.2604 \pm 0.0000$ | $0.9166 \pm 0.0000$ | $0.2800 \pm 0.0000$ |
| OCGIN | $0.6373 \pm 0.0276$ | $0.5650 \pm 0.2606$ | $0.6574 \pm 0.0487$ | $0.3208 \pm 0.0779$ | $0.6333 \pm 0.1261$ | $0.7387 \pm 0.1990$ |
| InfoGraph+DSVDD | $0.5137 \pm 0.0000$ | $0.6150 \pm 0.1594$ | $0.5519 \pm 0.1367$ | $0.7653 \pm 0.0806$ | $0.5417 \pm 0.2814$ | $0.3787 \pm 0.1049$ |
| GLocalKD | $0.6465 \pm 0.0066$ | $0.7063 \pm 0.1391$ | $0.2578 \pm 0.0000$ | $0.1979 \pm 0.0000$ | $0.8958 \pm 0.0335$ | $\mathbf{0.9719 \pm 0.0039}$ |
| OCGTL | $0.5394 \pm 0.0340$ | $0.6150 \pm 0.0903$ | $0.5009 \pm 0.0805$ | $0.6972 \pm 0.0939$ | $0.6792 \pm 0.0914$ | $0.9227 \pm 0.0116$ |
| DOHSC | $\mathbf{0.7784 \pm 0.0639}$ | $\mathbf{0.8600 \pm 0.0339}$ | $\mathbf{0.7601 \pm 0.1000}$ | $\mathbf{0.9181 \pm 0.0203}$ | $\mathbf{0.9583 \pm 0.0373}$ | $0.9653 \pm 0.0217$ |
| DO2HSC | $\mathbf{0.7928 \pm 0.0327}$ | $\mathbf{0.9050 \pm 0.0292}$ | $\mathbf{0.8443 \pm 0.0339}$ | $\mathbf{0.9375 \pm 0.0669}$ | $\mathbf{0.9792 \pm 0.0208}$ | $\mathbf{0.9800 \pm 0.0133}$ |

## M.2 GRAPH KERNELS AND GRAPH NEURAL NETWORKS

Graph kernels (Kriege et al., 2020) measure the similarity between graphs and are very useful in many tasks involving graphs, such as graph classification. A large body of work has emerged in the past years, including kernels based on neighborhood aggregation techniques and walks and paths. Shervashidze et al. (2011) introduced the Weisfeiler-Lehman (WL) algorithm, a well-known heuristic for graph isomorphism. In (Hido & Kashima, 2009), Neighborhood Hash kernel was introduced, where the neighborhood aggregation function is binary arithmetic. The most influential graph kernel for paths-based kernels is the shortest-path (SP) kernel (Borgwardt & Kriegel, 2005). For walks-based kernels, Gärtner et al. (2003) and Kashima et al. (2003) simultaneously proposed graph kernels based on random walks, which count the number of label sequences along walks that two graphs have in common. These graph kernel methods have the desirable property that they do not rely on the vector representation of data explicitly but access data only via the Gram matrix.

Another powerful and popular tool for handling graph data is the graph neural network. GNNs play a crucial role in effectively aggregating neighbor information for each node based on the edges in graph data. In the past decade, various improvements and enhancements for GNNs have been proposed (Welling & Kipf, 2016; Hamilton et al., 2017; Xu et al., 2019; Sun et al., 2020; Wu et al., 2023; Sun et al., 2023). GNNs can be applied to both node-level tasks and graph-level tasks. For graph-level tasks, one fundamental problem is graph representation learning, which aims to represent each graph as a vector and often requires a readout or pooling operation. Xu et al. (2019) showed that it is more effective to use a sum function to convert the representations of nodes of each graph to a vector, compared to mean and max functions. Wu et al. (2023) proposed a framework of graph learning based on kernel functions, which has a comparable or even better performance compared to GNNs.

## M.3 GRAPH-LEVEL ANOMALY DETECTION

There are few studies undertaken in graph-level anomaly detection (GAD). Existing solutions to GAD can be categorized into two families: two-stage and end-to-end. Two-stage GAD methods (Breunig et al., 2000; Schölkopf et al., 1999) first transform graphs into graph embeddings by graph neural networks or into similarities between graphs by graph kernels, and then apply off-the-shelf anomaly detectors. The drawbacks mainly include: 1) the graph feature extractor and outlier detector are independent; 2) some graph kernels produce "hand-crafted" features that are deterministic without much space to improve. Whereas, end-to-end approaches overcome these problems by utilizing deep graph learning techniques (such as graph convolutional network (Welling & Kipf, 2016) and graph isomorphism network (Xu et al., 2019)), which learn an effective graph representation while detecting graph anomaly (Zhao & Akoglu, 2021; Qiu et al., 2022; Ma et al., 2022).

In the past decades, regarding more end-to-end unsupervised graph-level anomaly detections, the graph kernel measures the similarity between graphs. It regards the result as a representation non-strictly or implicitly. However, the graph anomaly detection task associated with it usually performs a two-stage process, which cannot maintain the quality of representation learning while learning normal data patterns. Further concerning end-to-end models, Ma et al. (2022) proposed a global

and local knowledge distillation method for graph-level anomaly detection, which learns rich global and local normal pattern information by random joint distillation of graph and node representations while needing to train two graph convolutional networks jointly at a high time cost. Zhao & Akoglu (2021) combined the Deep Support Vector Data Description (Deep SVDD) objective function and graph isomorphism network to learn a hypersphere of normal samples. Qiu et al. (2022) sought a hypersphere decision boundary and optimized the representations learned by $k$ Graph Neural Networks (GNN) close to the reference GNN while maximizing the differences between $k$ GNNs, but did not consider the relationship between the graph-level representation and node features.

