# OpenReview forum: "Deep Orthogonal Hypersphere Compression for Anomaly Detection"
_ICLR.cc/2024/Conference — ICLR 2024 spotlight_

### Official Review · Reviewer_5jY6 · 2023-10-24

**Soundness:** 3 good
**Presentation:** 3 good
**Contribution:** 3 good
**Rating:** 8
**Confidence:** 3

**Summary:**

The paper suggests a new technique of hypersphere learning via a particular decision boundary to tackle the problem of Anomaly Detection. The boundary in this case is an orthogonal projection layer and the training data distribution is aligned with this geometry, a fact that encourages the correct prediction. The suggested methods seem to be ubiquitous in the data modalities, with emphasis on the graph data, and this is supported by numerical experiments.

**Strengths:**

- The paper is well written and the propositions seem sound.

- The paper solves the optimization problem (1) and this choice seems novel. The existing literature presented the following restrictions: (i) the decision surface inferred is not a standard hypersphere but at times a hyperellipsoid, leading to insufficient accuracy, (ii) the 'regular' data are located far from the hypersphere center, thus spoiling the normality of the predicted region and allowing anomalous data to fall into the sphere, (iii) the hypersphere is shows high sparsity leading to misclassification of the anomalous points.
The paper suggests (i) DOHC that employs an orthogonal projection layer that limits the evaluation errors, and (ii) DO2HSC that faces the second issue above using two co-centered hyperspheres.
The use of a regularization term in the objective function (1) avoids the correlation of the features and the problem of different variances.

- The authors provide extensive experiments in three different cases of datasets. Each case contains several datasets. The comparison to SOTA methods seems superior for the DOHSC and DO2HSC proposed model.

- The proposed architecture is applicable in graph data based on the optimization of (16) function.

**Weaknesses:**

Minimal weaknesses.
A weakness that can be pointed out is the use of one class in total for the anomaly detection problem, but this is clarified by the authors.
Good paper overall.

**Questions:**

- Can the authors offer some details about the averaging of scores (table 1)? Did they conduct repeatedly the same experiment like in tables 2,3?
- In the tabular data-based experiments, can the authors say why they chose F1 and not AUC ROC again, like in the graph data?
- For the visualization of the data, the paper pictures it 'by setting the projection dimension to 3'. Thus, was any further processing of the output applied? E.g. a dimensionality reduction technique? If yes, the AUC ROC score after this outcome may have changed from the table 3 score.

---

> ### Author Response · Authors · 2023-11-19
> **Rebuttal for Reviewer 5jY6**
>
> **Q1: Can the authors offer some details about the averaging of scores (table 1)? Did they conduct repeatedly the same experiment like in tables 2,3?**
>
> **Response:** Thank you for your inquiry. Similar to the approach detailed in Tables 2 and 3, we conducted each experiment 10 times to report the experimental results (the averages and standard deviations) in Table 1.
>
> **Q2: In the tabular data-based experiments, can the authors say why they chose F1 and not AUC ROC again, like in the graph data?**
>
> **Response:** Thank you for your question. In the tabular data-based experiments, we use F1-Score instead of AUC ROC in line with the evaluations used in many previous works such [1],[2],[3] to ensure the persuasiveness and fairness of the experiment. They were used as the baseline method in the experiment on tabular data, and we adopted the reported results from the original papers.
>
> [1] Liron Bergman and Yedid Hoshen. Classification-based anomaly detection for general data, ICLR 2020.
>
> [2] Sachin Goyal, Aditi Raghunathan, Moksh Jain, Harsha Vardhan Simhadri, and Prateek Jain. DROCC: Deep Robust One-Class Classification, ICML 2020.
>
> [3] Jinyu Cai and Jicong Fan. Perturbation learning based anomaly detection, NeurIPS 2022.
>
> **Q3: For the visualization of the data, the paper pictures it 'by setting the projection dimension to 3'. Thus, was any further processing of the output applied? E.g. a dimensionality reduction technique? If yes, the AUC ROC score after this outcome may have changed from the table 3 score.**
>
> **Response:** Thanks for the comment. To clarify, when visualizing the data by projecting it into a 3-dimensional space, we did not employ any additional dimensionality reduction techniques beyond setting the neuron number of the projection layer directly to 3. This means that the representation you see in the visualizations is a direct outcome of the neural network's projection layer, **without any further processing or transformation**.
>
> **Hope this response can solve your concerns. We thank the reviewer again for recognizing our work.**

---

### Official Review · Reviewer_7Ufo · 2023-10-27

**Soundness:** 3 good
**Presentation:** 3 good
**Contribution:** 3 good
**Rating:** 8
**Confidence:** 4

**Summary:**

This study centers on hypersphere-based anomaly detection challenges, presenting the orthogonal projection layer as an enhancement for deep SVDD. Additionally, the authors introduce the concept of bi-hypersphere anomaly detection. The effectiveness of these proposed modules is rigorously validated through a series of comprehensive experiments and insightful visualizations. Furthermore, the application of the two algorithms is extended to address graph-level anomaly detection, showcasing their versatility and potential impact in various contexts.

**Strengths:**

+ The paper's content is grounded in sound research, with a particularly innovative contribution in the form of the bi-hypersphere concept.

+ The research is substantiated by a comprehensive and diverse set of experiments, encompassing three distinct data types. The visualizations effectively convey the superiority of the proposed method.

+ The visualization results pertaining to anomaly detection are distinctive and intuitive, enhancing the paper's overall quality. The improvement over the previous baselines is remarkable.

**Weaknesses:**

- Why can the orthogonal projection layer ensure a standard hypersphere?

- The occurrence of the "soap bubble phenomenon" needs further clarification. Does it mean incompletely optimized?

- We know that Deep SVDD compels normal data close to the center of the decision boundary, why do anomalies appear within this decision boundary? What are the main differences and similarities between normal data and these anomaly data?

- Authors claimed that DO2HSC is to control training data to be more compact. I think the complete optimization of DOHSC can also reach this target, so what advantages does DO2HSC have about data compactness?

- Some details need to be double-checked, such as the bolding of three results in Table 13, Class 1 of MUTAG result, while the caption specifies marking only the best two results.

**Questions:**

See Strengths and Weaknesses.

---

> ### Author Response · Authors · 2023-11-19
> **Rebuttal for Reviewer 7Ufo**
>
> **W1: Why can the orthogonal projection layer ensure a standard hypersphere?**
>
> **Response:** Thank you for the comment. The orthogonal projection layer is designed to ensure that the latent representations are orthogonal, similar to the operation of Principal Component Analysis (PCA). By doing so, it reduces the correlation between different variables in the latent space and the variables have the same (unit) variance, which results in a more spherical distribution of samples. It was also demonstrated by our visualization experiment in the paper (see Fig. 12-15 in Appendix H). Note that ensuring uncorrelated (or even independent) and unit-variance variables (e.g. PCA, KPCA, and ICA) has shown effectiveness in novelty detection and statistical process control.
>
> **W2: The occurrence of the "soap bubble phenomenon" needs further clarification. Does it mean incompletely optimized?**
>
> **Response:** Thanks for reminding us to emphasize this aspect. The `soap bubble phenomenon' is not related to incomplete optimization. It is a common phenomenon of high-dimensional data. When we optimize the model, the sum of (squared) distances between all samples and the center is minimized, which is equivalent to maximum likelihood estimation (MLE), where the projected data points obey a Gaussian distribution and the mean vector is the center of the hypersphere. Even when the Gaussian assumption is realistic and the optimization is sufficient, the soap bubble phenomenon still exists due to our Proposition 1. Additionally, hypersphere compression can only ensure the sum of these distances is as small as possible, but it is challenging to guarantee that all distances are close to 0 according to the high-dimensional statistics. Thus, the 'soap bubble' is an actual issue that has been neglected by previous state-of-the-arts. We hope this explanation could address your concerns.
>
> **W3: We know that Deep SVDD compels normal data close to the center of the decision boundary, why do anomalies appear within this decision boundary? What are the main differences and similarities between normal data and these anomalous data?**
>
> **Response:** Thank you for highlighting this aspect. Deep SVDD assumes that the majority of the training data are normal instances and thus tries to minimize the volume of the hypersphere that encloses these normal data. Despite its design, anomalies can still appear within the decision boundary of Deep SVDD for several reasons: 1) Training data is insufficient or biased, so that the decision region is not compact enough for all normal data; 2) Anomalies that are similar to the normal data in certain dimensions might fall within the decision boundary. Consequently, an insufficiently compact training data distribution results in more empty holes in the decision region. These empty holes further pose a higher risk of anomalous data falling within them. The most remarkable difference is the different distributions followed by normal data and anomalous data. Even if precisely finding a perfect hypersphere decision region filled with normal data is challenging. However, the hypersphere contraction can isolate most of the abnormal data outside the decision boundary, and furthermore, the bi-hypersphere compression further enhances the compactness of the region. This is one of the motivations for proposing DO2HSC.
>
> In our paper, we provided an example of the anomalous data by Example 1. Suppose the input data is zero or very close to zero (or has many zero attributes), then after the projection of Deep SVDD, the data is at or close to the center of the hypersphere (origin). Although it is in the hypersphere, it is very different from the normal training data and should be treated as abnormal data. Deep SVDD failed to detect it. Real examples can be found in Figure 4.

---

> > ### Author Response · Authors · 2023-11-19
> > **Further Clarification**
> >
> > **W4: Authors claimed that DO2HSC is to control training data to be more compact. I think the complete optimization of DOHSC can also reach this target, so what advantages does DO2HSC have about data compactness?**
> >
> > **Response:** Thank you for bringing up this point. Firstly, the 'soap bubble' phenomenon poses a significant challenge for DOHSC,  as it tends to struggle in optimizing data compactness, especially in high-dimensional spaces where data are not uniformly distributed around the center of the hypersphere. On the other hand, DO2HSC is specifically designed to address this problem. The interval region defined by DO2HSC allows for a more compact and efficient representation of normal data, aligning better with the actual distribution patterns observed in high-dimensional datasets.
> >
> > Additionally, as mentioned in our response to **W2**, the 'soap bubble' issues are **not a result of incomplete optimizations**. They inherently exist in high-dimensional space, requiring a specifically designed algorithm to address them.
> >
> > Our Proposition 3 compared the compactness of DOHSC and DO2HSC. We can see that DO2HSC can provide a more compact decision region. By the way, the experimental results further demonstrated the superiority of DOH2SC over DOHSC.
> >
> > **W5: Some details need to be double-checked, such as the bolding of three results in Table 13, Class 1 of MUTAG result, while the caption specifies marking only the best two results.**
> >
> > **Response:** Thank you for pointing this out. We have revised these points and ensured consistency. We appreciate your attention to detail and your help in improving the accuracy and clarity of our research. The corrected version of Table 13 is updated in the revised manuscript.
> >
> > **Hope this response can solve your concerns. We thank the reviewer again for recognizing our work.**

---

### Official Review · Reviewer_mQBk · 2023-10-30

**Soundness:** 4 excellent
**Presentation:** 3 good
**Contribution:** 4 excellent
**Rating:** 8
**Confidence:** 5

**Summary:**

The paper studies the problem of unsupervised anomaly detection. The authors proposed a deep orthogonal hypersphere compression method, which has two variants. The authors also provided theoretical analysis. The experiments on images, tabular data, and graphs showed that the proposed methods are more effective than the competitors.

**Strengths:**

* The motivations and technique details of the proposed methods are clearly illustrated. The visualizations (e.g. Figures 1-4, 12 and 13) are very impressive.
* The ideas especially the two concentric hyperspheres for anomaly detection are novel and interesting. They provide new insights into anomaly detection.
* The theoretical analysis such as Propositions 1, 2, and 3 make the paper solid.
* The experiments are extensive. There are image datasets (e.g. CIFAR10), tabular datasets, and six graph datasets.
* More importantly, in the experiments, the proposed methods significantly outperformed state-of-the-art anomaly detection methods such as DROCC (2020), PLAD (2022), and GLocalKD (2022) and OCGTL (2022).

**Weaknesses:**

A minor weakness is that some points haven’t been sufficiently explained. Please refer to my questions.

**Questions:**

1. Figure 2 shows that the orthogonal projection improves the performance of anomaly detection. What is the fundamental reason? I suggest the authors provide further analysis as well as some references if possible.
2. A typo or grammar issue in the first paragraph of Section 2.1.2: ‘cannot be avoided by solving equation 1’, it is not an equation. It is an optimization problem.
3. Does Proposition 2 mean the distance (to the original or hypersphere center) based anomaly score in high-dimensional space are not reliable? If yes, the authors may add a few words to the last paragraph in Section 2.2.1 to provide a hint or motivation for the new anomaly score defined by (9).
4. Given Proposition 2, in the high-dimensional space, the normal data are already far away from the origin. Why do we need to further push them to the outer hypersphere, namely, performing the hypersphere compression to reduce the thickness of the shell?
5. Are $r_{max}$ and $r_{min}$ fixed or adjusted adaptively?
6. In Proposition 3, when $r_{min}=r_{max}$, $\kappa$ is infinity. Does this still make sense?
7. What make graph anomaly detection special compared to image and tabular data anomaly detection?
8. In Section 3.1, the authors mentioned a comparison method FCDD, but Table 2 doesn’t include the corresponding result.
9. In Appendix K, the authors showed the results of imbalanced experiments of graph data. Does it mean the results on graph data in the main paper are from balanced experiments? What is the difference between these two settings?

---

> ### Author Response · Authors · 2023-11-19
> **Rebuttal for Reviewer mQBk**
>
> **Q1: Figure 2 shows that the orthogonal projection improves the performance of anomaly detection. What is the fundamental reason? I suggest the authors provide further analysis as well as some references if possible.**
>
> **Response:** First, if there is no orthogonal project, we may encounter an inconsistency between the hypersphere assumption and the actual (optimal) decision boundary (e.g., an ellipsoid), which stems from the following two points: 1) the learned features have different variances and 2) the learned features are correlated. For instance, in the left plot of Fig. 2, the assumption and the anomaly score are based on the hypersphere but the actual decision boundary is an ellipsoid. Comparing the right plot with the left one reveals that the left plot has a lower true positive (TP) ratio  but a larger false positive (FP) ratio, which leads to a lower detection precision:
>
> $\frac{TP\downarrow}{TP\downarrow+FP\uparrow}$. Here we regarded normal data as the negative class and abnormal data as the positive class. Therefore, we propose to use the orthonormal projection. This refinement contributes to an improved performance. The idea is similar to that of applying PCA to fault detection in industrial processes [1].
>
> [1] T. Kourti and J. F. MacGregor, “Process analysis, monitoring and diagnosis, using multivariate projection methods,” Chemometrics Intell. Lab. Syst., vol. 28, no. 1, pp. 3–21, 1995.
>
> **Q2: A typo or grammar issue in the first paragraph of Section 2.1.2: ‘cannot be avoided by solving equation 1’, it is not an equation. It is an optimization problem.**
>
> **Response:** Thank you for pointing out this typo. We have revised it.
>
> **Q3: Does Proposition 2 mean the distance (to the original or hypersphere center) based anomaly score in high-dimensional space is not reliable? If yes, the authors may add a few words to the last paragraph in Section 2.2.1 to provide a hint or motivation for the new anomaly score defined by (9).**
>
> **Response:** We admit that the distance-based measures to the center (be it the original or hypersphere center) can be less reliable for anomaly detection. This is a consequence of the high-dimensional geometry and the behavior of distances in such spaces.
> We would add a statement that acknowledges the limitations of traditional distance-based anomaly scores in high-dimensional spaces, as highlighted by Proposition 2, and motivates the need for the new anomaly score. The added contents are shown as follows:
> "Given the implications of Proposition 2, we recognize that in high-dimensional spaces, traditional distance-to-center based anomaly scores may lose their reliability due to the concentration of measure phenomenon." This insight motivates our proposal of a new anomaly score as defined in Equation (9), which aims to address these high-dimensional challenges more effectively.
>
> **Q4: Given Proposition 2, in the high-dimensional space, the normal data are already far away from the origin. Why do we need to further push them to the outer hypersphere, namely, performing the hypersphere compression to reduce the thickness of the shell?**
>
> **Response:** We appreciate your concern. The primary goal of hypersphere compression is not just to push normal data further away from the center, but rather to reduce the thickness of the shell in which the normal data reside. This is particularly crucial in high-dimensional spaces where data points, including both normal and anomalous, tend to be equidistant from the center due to the curse of dimensionality.
>
> As Proposition 2 suggests, normal data are indeed farther away from the center as the dimensionality grows. However, the challenge lies in the relative positioning of normal and anomalous data. Without hypersphere compression, both normal and anomalous data might occupy a broad shell-like region, making it difficult to distinguish between them effectively.
>
> By employing hypersphere compression, we effectively reduce the volume of the space where normal data reside. This compression increases the contrast between normal and anomalous data, as anomalous data will now be more likely to fall outside this compressed hypersphere. This enhanced separation improves the sensitivity and specificity of the anomaly detection process.
>
> **Q5: Are $r_{max}$ and $r_{min}$ fixed or adjusted adaptively?**
>
> **Response:** $r_{\text{max}}$ and $r_{\text{min}}$ remain fixed throughout the entire training process after they are initialized by formula (7). This approach ensures that all normal data are optimized to lie within the interval area bounded by the bi-hyperspheres.

---

> > ### Author Response · Authors · 2023-11-19
> > **Further Clarification**
> >
> > **Q6: In Proposition 3, when $r_{\text{min}}=r_{\text{max}}$, $\kappa$ is infinity. Does this still make sense?**
> >
> > **Response:** Thanks for mentioning this point. In the case of $r_{\text{min}}=r_{\text{max}}$, the volume of DO2HSC is close to an infinitely thin shell, essentially transforming into the surface of a hypersphere. In this scenario, the data density of DO2HSC is obviously higher compared with that of DOHSC. However, it's important to note that this situation is quite rare, particularly in high-dimensional space. We have supplemented this special situation in Appendix D.
> >
> > **Q7: What make graph anomaly detection special compared to image and tabular data anomaly detection?**
> >
> > **Response:** Graph data, compared to other types of data, is inherently complex and rich in structural and relational information. Achieving a global representation for a graph involves harmonizing information across all nodes, while node representation further necessitates aggregating information from connected neighbors through edges. Consequently, it encompasses both independent node details and structural information. The crucial consideration is preserving a meaningful graph representation that is conducive to the anomaly detection task. This motivates our choice to maximize the mutual information among all nodes while simultaneously detecting anomalies.
> >
> > **Q8: In Section 3.1, the authors mentioned a comparison method FCDD, but Table 2 doesn’t include the corresponding result.**
> >
> > **Response:** Thanks for your careful review. We originally wanted to include FCDD as a baseline but found that FCDD uses additional data in the training, which means FCDD cannot be compared with our methods as well as the baselines considered in our paper. So we didn't include the results in Table 2. This is our neglect and we have corrected it in the revised paper.
> >
> > **Q9: In Appendix K, the authors showed the results of imbalanced experiments of graph data. Does it mean the results on graph data in the main paper are from balanced experiments? What is the difference between these two settings?**
> >
> > **Response:** The results on graph data in the main paper adhere to balanced settings. The data split involves allocating 80\% of the data from the normal class for training. The testing data is then constructed by combining the retained normal data with an equal or smaller number of anomalous data samples. Recognizing the significance of imbalance settings in practical anomaly detection scenarios, we follow the same data split for training and adjust the test dataset to a 10:1 ratio.
> >
> > **Hope this response can solve your concerns. We thank the reviewer again for recognizing our work.**

---

> > ### Comment · Reviewer_mQBk · 2023-11-23
> > **The response increases my confident**
> >
> > I appreciate the authors’ response, which increased my confidence in supporting the paper.

---

> > > ### Author Response · Authors · 2023-11-23
> > > **Gratitude for Confidence Boost**
> > >
> > > We appreciate your acknowledgment of the recent updates and responses presented. Your decision to raise the confidence level serves as a significant encouragement to our efforts. Thanks you once again for the constructive feedback, which has undoubtedly pushed our work forward.

---

### Official Review · Reviewer_wkqC · 2023-10-30

**Soundness:** 3 good
**Presentation:** 3 good
**Contribution:** 3 good
**Rating:** 8
**Confidence:** 5

**Summary:**

The paper introduces an innovative approach to anomaly detection, enhancing traditional hypersphere learning with an orthogonal projection layer. This improves accuracy and reduces false negatives. The paper also introduces a more compact decision region, a "hyperspherical shell," and extends the methods to graph-level anomaly detection. The experimental results demonstrate the effectiveness of these methods in comparison to existing approaches. The contributions include enhanced anomaly detection techniques, particularly beneficial for high-dimensional and graph-based data.

**Strengths:**

The paper stands out in the following aspects:
1.	Originality: The paper presents a problem that may lead to suboptimal performance in the field of anomaly detection and provides a solution, offering a novel approach to enhance the performance of anomaly detection algorithms.
2.	Quality: The research is of high quality, marked by a well-structured approach, rigorous validation, and superior performance compared to existing methods. The use of benchmark datasets adds to the credibility.
3.	Clarity: The paper is well written, ensuring clear communication of the research. It offers a logical flow.
4.	Significance: The paper addresses a novel anomaly detection issue, offering potential improvements for high-dimensional and graph-based data. The practical applicability and broad relevance make it highly valuable.

**Weaknesses:**

1.	This article mentions the concept of hyperspheres but doesn't provide a more rigorous theoretical explanation for why standard hyperspheres are superior to boundaries formed by ellipsoids. Adding mathematical proofs or a deeper theoretical foundation would strengthen the paper.
2.	High-dimensional data and large datasets pose scalability challenges. The paper could address the scalability of the proposed methods and discuss their efficiency and computational requirements in dealing with big data.

**Questions:**

1. The proof of Proposition 2 in section C of supplementary materials should be make more clear.

2. In equation 3, how to guarantee the projected embeddings is orthogonal via  singular value decomposition?

---

> ### Author Response · Authors · 2023-11-19
> **Rebuttal for Reviewer wkqC**
>
> **W1: This article mentions the concept of hyperspheres but doesn't provide a more rigorous theoretical explanation for why standard hyperspheres are superior to boundaries formed by ellipsoids. Adding mathematical proofs or a deeper theoretical foundation would strengthen the paper.**
>
> **Response:** Thank you for your insightful feedback. Actually, our paper does not involve the comparison between the standard hypersphere decision boundary and the ellipsoid decision boundary. We aim to address the inconsistency between the hypersphere assumption and the actual decision boundary of classical methods such as Deep SVDD. To be more precise, Deep SVDD assumes that, after the neural network transformation, the normal data are included in a standard hypersphere and use the distance of a data point to the center of the hypersphere as the anomaly score. However, the optimal decision boundary given by the neural network is often not a standard hypersphere, and can be an ellipsoid. The reason is that the optimization of Deep SVDD cannot ensure that the variables in the latent space are uncorrelated and have unit variance. The inconsistency between the computation of the anomaly score (assumption) and the actual optimal decision boundary (reality) degrades the detection accuracy. For a concrete example, please refer to Figure 2 in our paper. In the left plot, the black ellipse denotes the data distribution learnt by the encoder. However, hypersphere-based anomaly detection methods adopt the blue circle as the final evaluated decision boundary. This leads to more false positive samples (FP) and fewer true positive samples (TP), compared to the right plot, where we regarded normal data (purple) as the negative class and abnormal data (red) as the positive class. The detection precision, which is calculated as $\frac{TP\downarrow}{TP\downarrow+FP\uparrow}$, eventually decreases. We have incorporated these explanations into the revised manuscript to provide a more comprehensive understanding of our approach and its theoretical underpinnings.
>
> **W2: High-dimensional data and large datasets pose scalability challenges. The paper could address the scalability of the proposed methods and discuss their efficiency and computational requirements in dealing with big data.**
>
> **Response:** Thanks for your suggestion. Our models are trained via mini-batch optimization. Here we analyze the time and space complexity of our methods. Suppose the batch size is $b$, the maximum width of the hidden layers of the $L$-layer neural network is $w_{max}$, and the dimension of the input data is $d$, then the time complexities of the proposed methods are at most $\mathcal{O}(bdw_{max}LT)$, where $T$ is the total number of iterations. The space complexities are at most $\mathcal{O}(bd+dw_{max}+(L-1)w_{max}^2)$. We see that the complexities are linear with the number of samples, which means the proposed methods are scalable to large datasets. Particularly, for high-dimensional data (very large $d$), we can use small $w_{max}$ to improve the efficiency.
>
> **Q1: The proof of Proposition 2 in section C of supplementary materials should be made more clear.**
>
> **Response:** Thanks for your interest in the proof of Proposition 2. In the proof for Proposition 2, we start by leveraging the result from Proposition 1 and the properties of the function $f$, which is $\eta$-Lipschitz. The central idea is to connect the bounds on the norm of $\mathbf{s}-\bar{\mathbf{c}}$ with those on $\mathbf{z}-\mathbf{c}$, using the Lipschitz condition. We have added a step-by-step explanation to the manuscript, which aids in making the logical flow more apparent.
>
> **Q2: In equation 3, how to guarantee the projected embeddings is orthogonal via singular value decomposition?**
>
> **Response:** Thanks for mentioning this point. The singular value decomposition can be formulated
> $$\mathbf{U}\mathbf{\Lambda}\mathbf{V}^\top =\mathbf{Z},$$
> where $\mathbf{U}$ contains a set of vectors orthogonal to each other. In our paper, the projection matrix is $\mathbf{W}=\mathbf{V}\_{k'} \mathbf{\Lambda}\_{k'}^{-1}$, where $\mathbf{V}\_{k'}$ consists of the first $k'$ columns of $\mathbf{V}$. It follows that $\tilde{\mathbf{Z}}=\mathbf{Z}\mathbf{W}=\mathbf{U\Lambda V}^\top\mathbf{V}\_{k'} \mathbf{\Lambda}\_{k'}^{-1}=\mathbf{U}\_{k'}$, where $\mathbf{U}\_{k'}$ is composed of the first $k'$ columns of $\mathbf{U}$. Therefore, the columns of the embeddings $\tilde{\mathbf{Z}}$ are orthogonal naturally, namely $\tilde{\mathbf{Z}}^\top\tilde{\mathbf{Z}}=\mathbf{U}\_{k'}^\top\mathbf{U}\_{k'}=\mathbf{I}\_{k'}$.
>
> **Hope this response can solve your concerns. We thank the reviewer again for recognizing our work.**

---

### Meta-Review · Area_Chair_1dqN · 2023-12-03

**Metareview:**

All the reviewers reach a consensus that the paper should be accepted, and I also find that the paper has a good shape. Hence, I would recommend accepting the paper.

**Justification For Why Not Higher Score:**

Anomaly detection is a relatively narrow topic.

**Justification For Why Not Lower Score:**

N/A

---

### Decision · Program_Chairs · 2024-01-16

Accept (spotlight)